# Mapping brain-wide excitatory projectome of primate prefrontal cortex at submicron resolution and comparison with diffusion tractography

**Mingchao Yan[1,2†], Wenwen Yu[3†], Qian Lv[4], Qiming Lv[1], Tingting Bo[1,2], Xiaoyu Chen[1,2], Yilin Liu[1,2], Yafeng Zhan[1,2], Shengyao Yan[1,2], Xiangyu Shen[1], Baofeng Yang[3], Qiming Hu[5], Jiangli Yu[5], Zilong Qiu[1], Yuanjing Feng[5], Xiao-Yong Zhang[3], He Wang[3], Fuqiang Xu[6], Zheng Wang[4]\***

[1]Institute of Neuroscience, State Key Laboratory of Neuroscience, Center for Excellence in Brain Science and Intelligence Technology, Chinese Academy of Sciences, Shanghai, China; [2]University of Chinese Academy of Sciences, Beijing, China; [3]Institute of Science and Technology for Brain-inspired Intelligence, Fudan University, Shanghai, China; [4]School of Psychological and Cognitive Sciences; Beijing Key Laboratory of Behavior and Mental Health; IDG/McGovern Institute for Brain Research; Peking-Tsinghua Center for Life Sciences, Peking University, Beijing, China; [5]College of Information Engineering, Zhejiang University of Technology, Hangzhou, China; [6]Shenzhen Key Lab of Neuropsychiatric Modulation and Collaborative Innovation Center for Brain Science, Guangdong Provincial Key Laboratory of Brain Connectome and Behavior, CAS Key Laboratory of Brain Connectome and Manipulation, Brain Cognition and Brain Disease Institute (BCBDI), Shenzhen Institutes of Advanced Technology, Shenzhen-Hong Kong Institute of Brain Science-Shenzhen Fundamental Research Institutions, Shenzhen, China

**\*For correspondence:**
zheng.wang@pku.edu.cn

[†]These authors contributed equally to this work

**Competing interest:** The authors declare that no competing interests exist.

**Abstract** Resolving trajectories of axonal pathways in the primate prefrontal cortex remains crucial to gain insights into higher-order processes of cognition and emotion, which requires a comprehensive map of axonal projections linking demarcated subdivisions of prefrontal cortex and the rest of brain. Here, we report a mesoscale excitatory projectome issued from the ventrolateral prefrontal cortex (vlPFC) to the entire macaque brain by using viral-based genetic axonal tracing in tandem with high-throughput serial two-photon tomography, which demonstrated prominent monosynaptic projections to other prefrontal areas, temporal, limbic, and subcortical areas, relatively weak projections to parietal and insular regions but no projections directly to the occipital lobe. In a common 3D space, we quantitatively validated an atlas of diffusion tractography-derived vlPFC connections with correlative green fluorescent protein-labeled axonal tracing, and observed generally good agreement except a major difference in the posterior projections of inferior fronto-occipital fasciculus. These findings raise an intriguing question as to how neural information passes along long-range association fiber bundles in macaque brains, and call for the caution of using diffusion tractography to map the wiring diagram of brain circuits.

## Editor's evaluation

This paper uses a novel technique in combination with high throughput microscopy to generate a detailed map of macaque prefrontal connections. It will not only be of interest to anatomists, but

also to the neuroimaging community, as it includes a detailed comparison to MRI-based connectivity approaches commonly used to study the human brain.

## Introduction

Higher-order processes of cognition and emotion regulation that depend on the prefrontal cortex are all based on multiple, long-range connections between neurons (*Neubert et al., 2014*; *Carlén, 2017*; *Borra et al., 2011*). Axons connecting local and distant neurons form a fundamental skeleton of the brain circuitry, which is of paramount importance to fathom the organization of in-/output pathways that enable those vital functions (*Oh et al., 2014*; *Zingg et al., 2014*). Given the complexity and heterogeneity of the primate prefrontal cortex (*Carlén, 2017*), understanding the working mechanisms of the prefrontal cortex requires a comprehensive map of axonal projections linking its demarcated subdivisions and the rest of brain. A subdivision of the prefrontal cortex - the ventrolateral section (vlPFC), which mainly spans Brodmann's Areas 44, 45 a/b, 46 v/f, and 12 r/l (*Saleem et al., 2014*), is central to a variety of functions including language, objective memory, and decision-making (*Sakagami and Pan, 2007*; *Levy and Wagner, 2011*). Emerging evidence further demonstrates abnormalities of vlPFC in tight association with deficits in cognitive flexibility (*Neubert et al., 2014*; *Cai et al., 2020*; *Zhan et al., 2021*), suggesting that an elaborate delineation of its hard wiring would shed light on the underlying neuropathology of psychiatric disorders (*Haber et al., 2020*).

Such neuroanatomical inter-areal connectivity has been probed using invasive bulk injections of tracers and noninvasive imaging methods with millimeter-scale spatial resolution (*Zeng, 2018*; *Glasser et al., 2016*; *Wang et al., 2013*). Histological neural tracing has been historically utilized for circuit/pathway mapping and continues to be the most reliable way of survey for all myelinated axons in mammalian brains (*Zeng, 2018*; *Luo et al., 2018*; *Bienkowski et al., 2018*), which has also been used as a gold standard to validate other modalities like diffusion tractography (*Schmahmann et al., 2007*; *Donahue et al., 2016*; *Reveley et al., 2015*; *Mortazavi et al., 2018*; *Folloni et al., 2019*; *Dyrby et al., 2018*). Diffusion tractography, which has been developed in the 1990s to estimate the tissue microstructure by means of spatial encoding of water molecule movements (*Basser et al., 1994*), represents the only methodology capable of inferring the ensemble of anatomical connections in the living animal or human brain (*Wedeen et al., 2012*; *Mori and Zhang, 2006*). But this technique is an indirect observation with limited resolution and accuracy, and its reliability of false negative and false positive findings remains to be fully validated in a 3D space (*Donahue et al., 2016*; *Maier-Hein et al., 2017*). Notably, some classic tract-tracing methods are not sensitive to specific neuronal types or axonal trajectories. They do not report the traveling course in a 3D space through which the axons travel for a remarkably long distance (i.e. over centimeter length). The pursuit of long-range axonal fiber tracing across the entire monkey brain has become feasible thanks to rapid advances in viral and genetic tools in the primate species, tissue labeling, large-scale microscopy, and computational image analysis (*Han et al., 2009*; *Nassi et al., 2015*; *Albanese and Chung, 2016*; *Bedbrook et al., 2018*). Moreover, viral-based techniques for targeting specific neuronal types in macaque brain have achieved remarkable success (*Han et al., 2009*; *Stauffer et al., 2016*), which may furnish the requisite biological detail including excitatory and inhibitory in-/output to enrich structural network reconstructions for improved prediction of brain function (*Suárez et al., 2020*). However, it remains unclear thus far what type of viral vector is suitable for long-range axonal fiber tracing (*Zeng, 2018*; *Luo et al., 2018*).

Cross-comparison of the fiber details generated by two modalities with spatial scale differences in order of magnitude is technically demanding as many cellular structures or fiber pathways of biological interest are rather small relative to the voxel size of most diffusion MRI data (*Mori and Zhang, 2006*). One of the challenging undertakings is to image long-range axonal fibers of many neurons with sufficiently high resolution to enable tracking axonal trajectories across the entire brain (*Ragan et al., 2012*; *Li et al., 2010*), which has stirred some debates such as right-angle fiber crossings (*Mortazavi et al., 2018*; *Wedeen et al., 2012*) and the existence of the inferior fronto-occipital fasciculus (IFOF) in the primate brain (*Forkel et al., 2014*). The IFOF first proposed in the early 19th century supposedly connects the ventrolateral prefrontal cortex and medial orbitofrontal cortex to the occipital lobe through the ventral part of the external capsule (*Curran, 1909*; *Catani et al., 2002*). Micro-dissection and diffusion MRI tractography studies have recently confirmed the anatomy of this

**eLife digest** In the brain is a web of interconnected nerve cells that send messages to one another via spindly projections called axons. These axons join together at junctions called synapses to create circuits of nerve cells which connect neighboring or distant brain regions. Notably, long-range neural connections underpin higher-order cognitive skills (such as planning and emotion regulation) which make humans distinct from our primate relatives. Only by untangling these far-reaching networks can researchers begin to delineate what sets the human brain apart from other species.

Researchers deploy a range of imaging techniques to map neural networks: scanning entire brains using MRI machines, or imaging thin slices of fluorescently labelled brain tissue using powerful microscopes. However, tracing long-range axons at a high resolution is challenging, and has stirred up debate about whether some neural tracts, such as the inferior fronto-occipital fasciculus, are present in all primates or only humans.

To address these discrepancies, Yan, Yu et al. employed a two-pronged approach to map neural circuits in the brains of macaques. First, two techniques – called viral tracing and two-photon microscopy – were used to create a three-dimensional, fine-grain map showing how the ventrolateral prefrontal cortex (vlPFC), which regulates complex behaviors, connects to the rest of the brain. This revealed prominent axons from the vlPFC projecting via a single synapse to distant brain regions involved in higher-order functions, such as encoding memories and processing emotion. However, there were no direct, monosynaptic connections between the vlPFC and the occipital lobe, the brain's visual processing center at the back of the head.

Next, Yan, Yu et al. used a specialized MRI scanner to create an atlas of neural circuits connected to the vlPFC, and compared these results to a technique tracing axons stained with a fluorescent dye. In general, there was good agreement between the two methods, except for major differences in the rear-end projections that typically form the inferior fronto-occipital fasciculus. This suggests that this long-range neural pathway exists in monkeys, but it connects via multiple synapses instead of a single junction as was previously thought.

The findings of Yan, Yu et al. provide new insights on the far-reaching neural pathways connecting distant parts of the macaque brain. It also suggests that atlases of neural circuits from whole brain scans should be taken with caution and validated using neural tracing experiments.

pathway (*Sarubbo et al., 2019*; *Barrett et al., 2020*). Despite an abundance of functional evidence supporting a central role of occipito-frontal circuitry in cognition and sensory integration, a number of axonal tracing studies, which have been able to identify monosynaptic connections, have failed to reveal the IFOF in the macaque brain (*Schmahmann et al., 2007*; *Schmahmann and Pandya, 2006*). By contrast, sparse connections between frontal and occipital cortices in macaques were reported by other labs (*Gerbella et al., 2010*; *Markov et al., 2014*), although they do not show whether these axons follow the course expected for the IFOF. As such, a detailed anatomical definition of the IFOF is a topic of active research (*Barrett et al., 2020*).

In the present study, we aim to establish a comprehensive brain-wide excitatory projectome of the vlPFC in macaque monkeys using viral-based genetic tracing in tandem with serial two-photon (STP) tomography (*Figure 1A*), a technique that has successfully achieved high-throughput fluorescence imaging of the entire mouse brain by integrating two-photon microscopy and tissue sectioning (*Ragan et al., 2012*). We also performed ex vivo dMRI scans of the entire macaque brains to track the axonal fiber tracts using an ultra-high field (11.7T) MRI scanner (*Figure 1C*). In addition, in a common 3D space (*Figure 1B*), we intended to make a direct comparison of the mesoscale projectome to that derived from ultra-high field diffusion tractography (*Figure 1C and D*).

## Results

### Determination of viral vectors for long-range anterograde tracing

We tested whether VSV, lentivirus, and AAV vectors with demonstrated success in rodents worked in the macaque brain and which vector was best suitable for long-range axonal fiber tracing. Five days after infection with VSV-ΔG, the neuronal cell bodies in the cerebral cortex (*Figure 2A and B*)

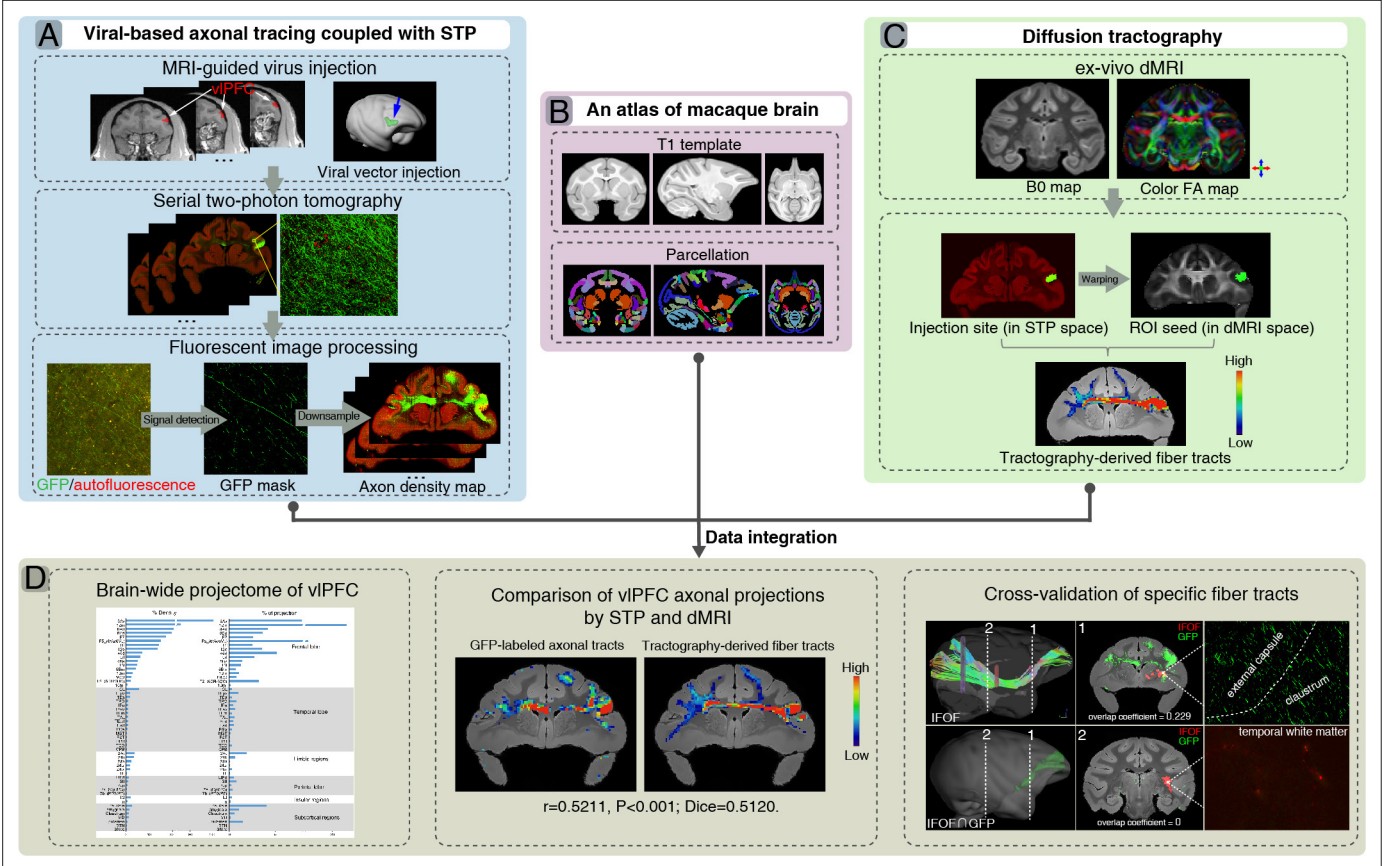

**Figure 1.** A flowchart diagram for brain-wide analyses of ventrolateral prefrontal cortex (vlPFC) projectome in macaques. The pipeline integrates the serial two-photon (STP) data in the mesoscopic domain (**A**) with macroscopic dMRI data (**C**) in a common 3D space (**B**). (**A**) T1 images were used to guide stereotaxic injection of AAV vectors to vlPFC (upper panel). High-throughput fluorescent images of viral-based genetic axonal tracing were acquired by STP tomography throughout the brain, which enables a close-up view and quantitative analysis of any region of interest (middle panel). A supervised machine learning approach was used for segmentation of GFP-positive signal and removal of autofluorescence in STP data. The serial segmented GFP images were down-sampled to compute the total signal intensity for each 200 μm × 200 μm grid by summing the number of signal-positive pixels in that grid and to generate the axonal density map (bottom panel). (**B**) An MRI atlas of cynomolgus macaques was used to construct a common 3D space. (**C**) Ex-vivo dMRI of macaque brain was acquired using an 11.7T MRI scanner, illustrated as representative B0 (left) and direction-encoded color FA maps (right). Using the injection site identified from the STP data as seed regions, the target fiber tracts can be derived from diffusion tractography. (**D**) Integration of STP and dMRI data was implemented in a common 3D space, which allows quantitative analyses including whole-brain analysis of axonal projectome (left), comparison of vlPFC projectome by STP and dMRI (middle), and cross-validation of fiber tracking in both STP and dMRI (right).

The online version of this article includes the following figure supplement(s) for figure 1:

**Figure supplement 1.** A customized setup of serial two-photon tomography for macaque brain.

**Figure supplement 2.** A typical STP tomography image set of a single macaque brain.

**Figure supplement 3.** Autofluorescence in macaque brain.

**Figure supplement 4.** Difference between axonal varicosities and dot-looking background.

**Figure supplement 5.** Co-registration of STP images to the MRI-based template of macaque brain.

**Figure supplement 6.** Representative fluorescent images showing injection site and major tracts of sample #7.

and mediodorsal (MD) thalamus (*Figure 2—figure supplement 1A*) were clearly labeled with GFP, although only proximal neurites were labeled with no long-range axonal fibers detected (*Figure 2—figure supplement 2*). When the infection time was extended to about a month, we observed widespread axon loss and neuronal cell death (*Figure 2—figure supplement 1D-G*). The long-term-infected neurons underwent morphological changes such as membrane blebbing (*Figure 2—figure supplement 1B and C*), a key morphological change associated with the induction of apoptosis. Local

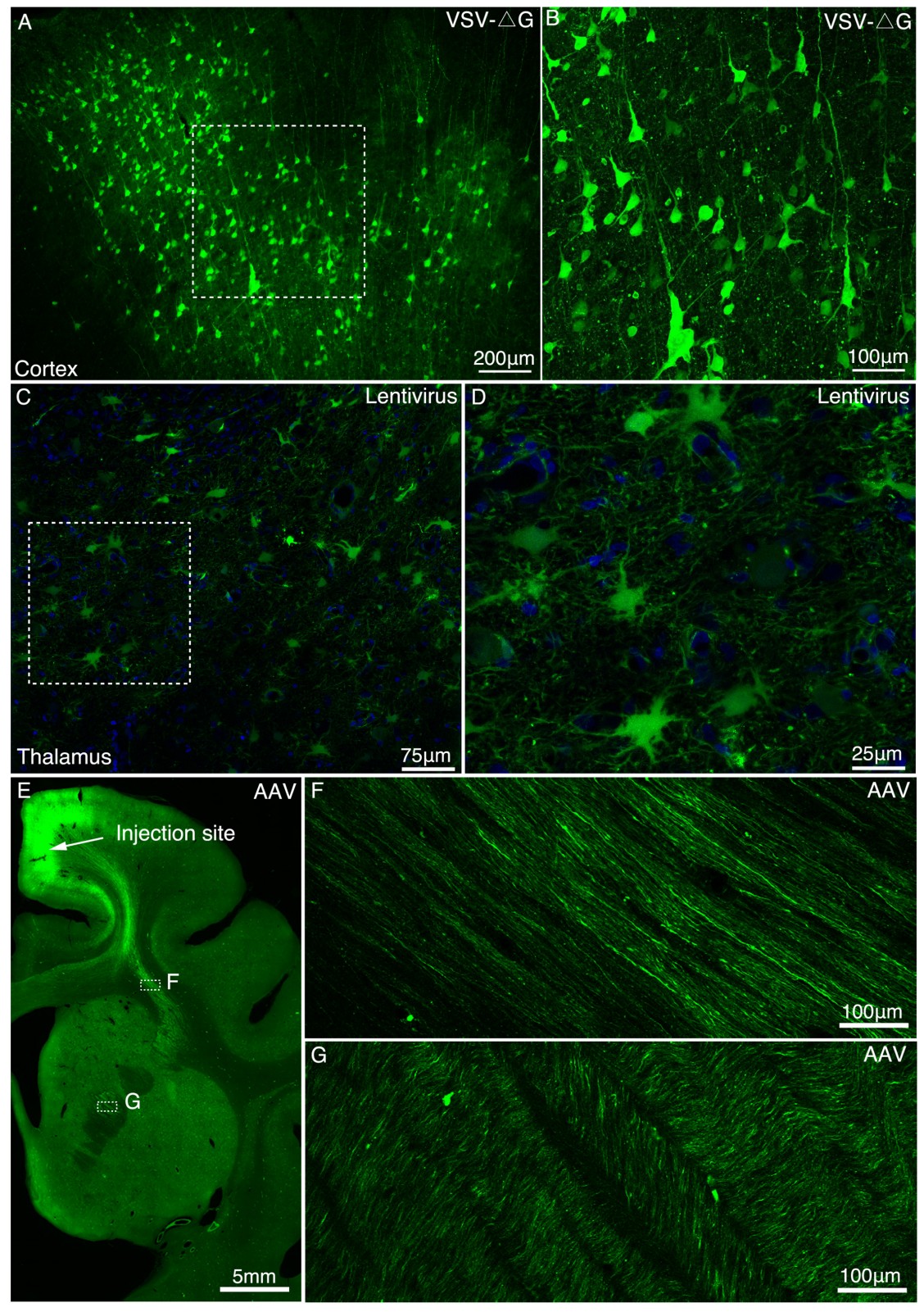

**Figure 2.** Determination of viral vectors for long-range anterograde tracing in macaques. (**A**) GFP-labeled neurons were found in the premotor cortex ~5 days after injection of VSV-ΔG encoding Tau-GFP. (**B**) A magnified view illustrating the morphology of GFP-labeled neurons in the area outlined with a white box in (A). (**C**) Lentivirus construct was injected into the macaque thalamus and examined for transgene expression after ~9 months. (**D**) High power views of the dotted rectangle in panel C. (**E**) GFP-labeled neurons and axons were observed in the premotor cortex ~45 days after injection of

*Figure 2 continued on next page*

*Figure 2 continued*

AAV2/9 encoding Tau-GFP. Two dashed line boxes enclose the regions of interest: frontal white matter and ALIC, whose GFP signal are magnified in (**F**) and (**G**), respectively.

The online version of this article includes the following figure supplement(s) for figure 2:

**Figure supplement 1.** Long-term expression of VSV-△G induced neurotoxicity in macaque brain.

**Figure supplement 2.** Expression of GFP using VSV-△G injected into the MD thalamus of macaque brain.

**Figure supplement 3.** Expression of GFP using lentivirus injected into the MD thalamus of macaque brain.

**Figure supplement 4.** Expression of GFP using AAV2/9 injected into the MD thalamus of macaque brain.

**Figure supplement 5.** AAV-infected cells were stained for NeuN and GFAP antibodies.

**Figure supplement 6.** Comparison of two AAV constructs.

**Figure supplement 7.** Long-range axonal tracing outcomes using AAV2/9, lentivirus,` and VSV-△G.

injection with VSV-ΔG mediated rapid and transient gene expression nearby the injection site, and an extension of infection time evidently caused fatal neurotoxicity.

Lenti-Ubic-GFP exhibited stable expression in the cell soma even after 9 months (*Figure 2C and D*), despite sparse labeling of GFP positive axons (*Figure 2—figure supplement 3*). By contrast, 6 weeks after AAV2/9-CaMKIIa-Tau-GFP was injected into the premotor cortex (*Figure 2E-G*) or MD thalamus (*Figure 2—figure supplement 4*), axonal fiber bundles like anterior limb of internal capsule (ALIC) (*Figure 2G*) were clearly visualized over several centimeters. The AAV-infected cells were positive for the neuronal specific marker NeuN, but negative for astrocyte specific marker GFAP (*Figure 2—figure supplement 5*). As a validation test, AAV2/9 construct encoding mCherry was co-injected with AAV2/9 construct encoding Tau-GFP into the premotor cortex. And we found that the signal intensity of most Tau-GFP labeled axons was consistently higher than that of mCherry labeled axons (*Figure 2—figure supplement 6A-D*).

We compared the axonal fiber tracing efficiency of VSV-ΔG, Lentivirus and AAV2/9 (AAV2/9-CaMKIIa-Tau-GFP) in the MD thalamus (*Figure 2—figure supplement 7*). The density of axonal fibers labeled by AAV2/9 (*Figure 2—figure supplement 7C*) was significantly higher ($P < 0.001$, *Figure 2—figure supplement 7D*) than by Lentivirus (*Figure 2—figure supplement 7B*) and VSV-ΔG (*Figure 2—figure supplement 7A*). VSV-ΔG labeled axons sparse in the proximal, and the axonal density decreased sharply (*Figure 2—figure supplement 7D*). Axons labeled by lentivirus (*Figure 2—figure supplement 7B*) were also significantly denser ($P < 0.01$, *Figure 2—figure supplement 7D*) than by VSV-ΔG (*Figure 2—figure supplement 7A*) distant from the injection site.

## Brain-wide excitatory projectome of vlPFC in macaques

AAV2/9 encoding Tau-GFP under the control of excitatory promoter CaMKIIa was determined as an anterograde tracer for mapping the excitatory projectome of vlPFC. The injection site in vlPFC, validated by STP images, including area 45, 12 l, and 44, was precisely located in cortical gray matter (Figure 4A-D). To identify the cell type specificity of Tau-GFP gene expression driven by the CaMKII-α promoter, immunofluorescent staining was performed with antibodies against the excitatory neuron specific marker CaMKII-α and the inhibitory neuron-specific neurotransmitter GABA. GFP-positive neurons in the injection site were observed positive for CaMkIIa (*Figure 4—figure supplement 1A-C*) and negative for GABA (*Figure 4—figure supplement 1D-F*), indicating that the AAV labeled neurons were glutamate excitatory neurons.

To acquire a detailed account of the brain-wide vlPFC projectome, we analyzed its connectivity profile with other 173 parcellated regions in the monkey brain atlas using the STP tomography data (*Figures 3 and 4*). The GFP-labeled projecting axons largely encompassed the anterior part of the brain including the frontal lobe, temporal lobe, limbic regions, insular, and some subcortical regions, but no labeled axons were found in the occipital lobe (*Figure 3A–C*, *Figure 3—figure supplement 1*). Within the frontal lobe, GFP-labeled projecting axons were markedly dense in the OFC, rostrally distributed in area 12 m (*Figure 4E, F and G*), 12o (*Figure 4E*), 11 l (*Figure 4E, H and I*), 13 l (*Figure 4E*), and 13 m (*Figure 4E*). The 12 m received strongest innervation from vlPFC relative to other OFC subregions (*Figure 4E*). Laterally, axonal projections were found in the FEF including 8Av (*Figure 4—figure supplement 2A, B and C*) and 8Ad (*Figure 4—figure supplement 2A and D*).

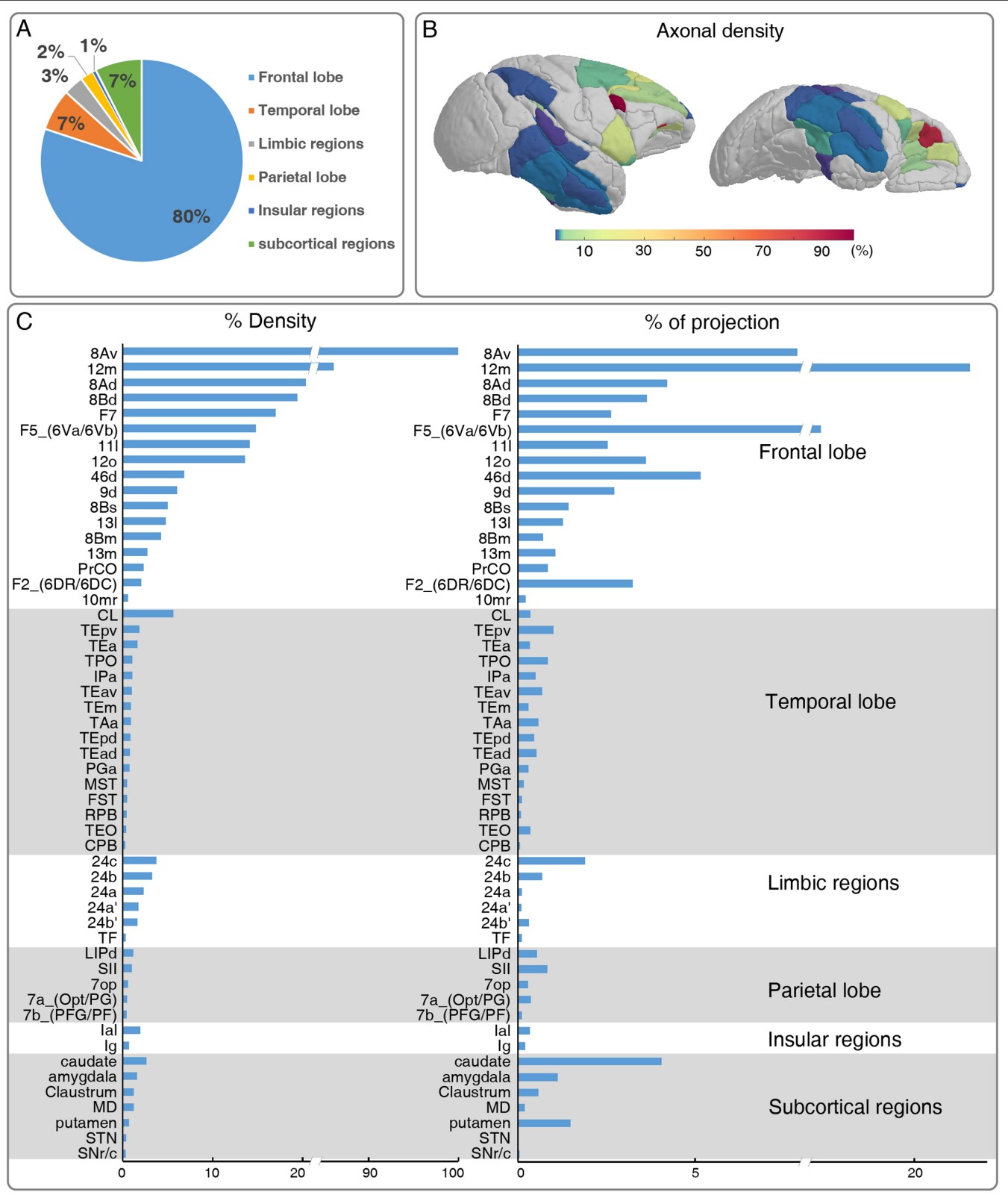

**Figure 3.** Brain-wide distribution of GFP-labeled excitatory projectome of ventrolateral prefrontal cortex (vlPFC). (**A**) A pie chart shows the brain-wide distribution of vlPFC axonal projectome. (**B**) The normalized percentage distribution of axonal density was rendered onto a 3D brain surface. (**C**) The histogram plots show the vlPFC projections to other regions where the connectivity strength was quantified by the density of GFP-positive axons and proportion of total projection. We calculated the innervation density, given in percent of strongest projection.

*Figure 3 continued on next page*

*Figure 3 continued*

The online version of this article includes the following source data and figure supplement(s) for figure 3:

**Source data 1.** Entire dataset for the brain-wide distribution of vlPFC axonal projectome used in *Figure 3*.

**Figure supplement 1.** Drawings of coronal sections showing the distribution of the anterograde viral labeling after the injection in vlPFC (Case #8).

Dorsally, there were dense axonal fibers in the dorsal prefrontal cortex, including area 8Bd (*Figure 4—figure supplement 2E, F and G*), 8Bm (*Figure 4—figure supplement 2E*), 8Bs (*Figure 4—figure supplement 2E*), 46d (*Figure 4—figure supplement 2E, H and I*), and 9d (*Figure 4—figure supplement 2E, J and K*). The 8Bd and 46d received relatively more innervation compared with 9d, 8Bs, and 8Bm (*Figure 4—figure supplement 2E*). On the medial surface of the brain, scattered axon fibers were visible in area F5 (*Figure 4—figure supplement 2L and M*), F7 (*Figure 4—figure supplement 2L and N*), and F2 (*Figure 4—figure supplement 2L*) of the premotor cortex. The axons with the premotor cortex exhibited a gradient pattern with the largest axon distribution along the anterior part (*Figure 4—figure supplement 2L*). In addition, axons were noted in the precentral opercular area (PrCO) and medial prefrontal area (mainly in 10mr) (*Figure 3C*). Interestingly, the projections anchored in the prefrontal cortex of these axonal fibers formed isolated clusters (*Figure 4F and H*, indicated by arrows). The z-axis extent of axonal clusters was ranging from 1.2 mm to 3.8 mm (2.24 ± 0.80 mm) (*Figure 4—figure supplement 3*).

Beyond the frontal lobe, rich connections were observed in the temporal lobe (*Figure 3*), predominantly in caudal lateral (CL), caudal (CPB), and rostral (RPB) portions of parabelt region of the auditory cortex; anterior TE (TEa), medial TE (TEm), superior temporal polysensory area (STP, correspond to areas PGa and TPO), IPa and TAa of the dorsal bank/ventral bank/fundus of the superior temporal sulcus (STSd/v/f) (*Figure 3C*); medial superior temporal area (MST), floor of superior temporal area (FST), anteroventral TE (TEav), anterodorsal TE (TEad), and area TEO of superior temporal area. The vlPFC also sends axons to limbic regions, mainly in 24 a, 24 a', 24b, 24b', and 24 c of anterior cingulate areas (ACC) (*Figure 3C*); area TF of parahippocampal cortex (*Figure 3C*). Relatively weak projections were observed in the dorsal subdivision of lateral intraparietal area (LIPd); 7 a and 7b of inferior parietal lobule areas; secondary somatosensory area (SII) and parietal operculum (7op) of the parietal cortex (*Figure 3C*). There were some sparsely labeled axons in the granular insula (Ig) and lateral agranular insula (Ial) area (*Figure 3C*). In white matter, traveling axonal bundles were found in the corpus callosum, anterior limb of internal capsule (ALIC, *Figure 4—figure supplement 4A, B*), and anchored into the MD thalamus (*Figure 4—figure supplement 4C, D*). Subcortically, axon clusters were observed in the medial (*Figure 4—figure supplement 4F, G*) and caudal (*Figure 4—figure supplement 4H*) parts of caudate. High resolution confocal images revealed that axons in MD (*Figure 4—figure supplement 4D*) and caudate (*Figure 4—figure supplement 4J*) were thinner than those in the ALIC (*Figure 4—figure supplement 4B*). Furthermore, the labeled axons were found extending to the parvicellular part of accessory basal nucleus of amygdala (ABpc), reticulate and compacta parts of substantia nigra (SNr/c), claustrum and subthalamic nucleus (STN) (*Figure 3C*).

## Comparison of vlPFC axonal projections by dMRI and STP

We further introduced a quantitative comparison of vlPFC connectivity profile obtained by dMRI-based tractography and STP data. Typical T2-weighted and dMRI images of the macaque brain acquired from an ultra-high field MRI scanner were shown in *Figure 5A–D*. GFP projection and probabilistic tract were plotted with the Dice coefficients and Pearson coefficients (R) along the anterior-posterior axis of the whole macaque brain. The Dice coefficients and Pearson coefficients were higher in dense projection regions, especially for the vlPFC-CC-contralateral tract (*Figure 6A*). To carry out a proof-of-principle investigation, we focused on the vlPFC-CC-contralateral tract that was reconstructed in 3D space by using STP and dMRI data, respectively (*Figure 6B and C*). After co-registering the reconstructed tracts into a common 3D space, our approach relied on slice-based statistical correlation methods (the Pearson correlation and Dice coefficients) along this vlPFC-CC-contralateral tract. Upon visual comparison, the dMRI-derived tracts largely overlapped with the axonal bundles shown in STP images (*Figure 6B and C*). Statistical correlation indices were computed for each pair of diffusion tractography and STP images to quantify their spatial overlap. We found consistent, marked agreement between these two modalities along this tract, as demonstrated in *Figure 6D–F*. For all slices (spaced

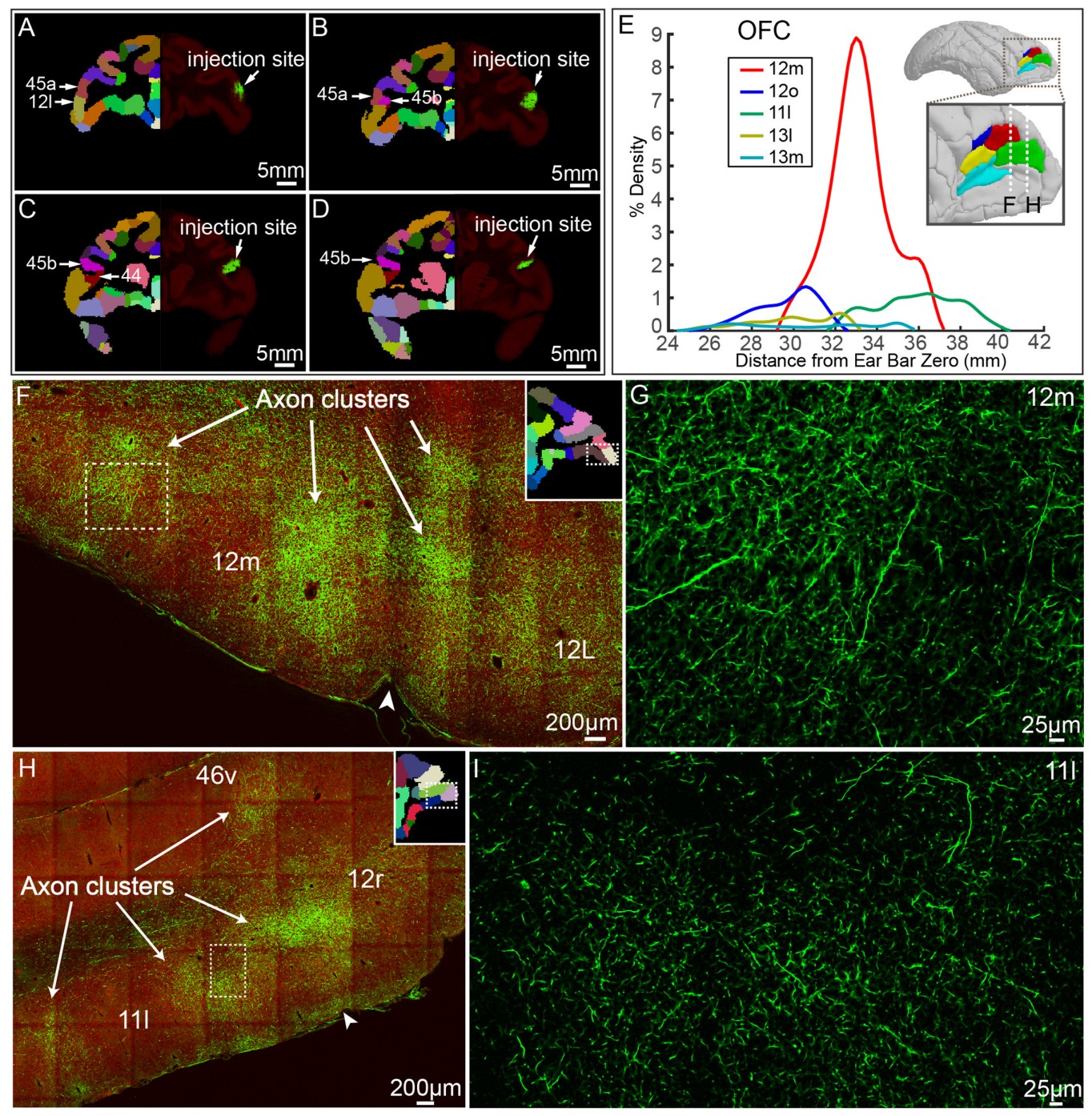

**Figure 4.** ventrolateral prefrontal cortex (vlPFC) projectome within the prefrontal lobe. (**A–D**) Representative coronal slices of the injection site in vlPFC are shown overlaid with the monkey brain template (left hand side), mainly spanning areas 45 a, 45b, 12 l, and 44. (**E**) Percentage of output density of vlPFC projectome along the anterior-posterior axis of the OFC. The inset shows the spatial location of individual Broadmann areas in OFC. Dotted lines indicate anterior-posterior position of the following fluorescent images. (**F–I**) Representative two-photon images of vlPFC axonal projections to OFC: 12 m and 11 l. Arrows indicate the axon clusters. Insets show the low power images of the section indicating the position of the higher power images.

The online version of this article includes the following figure supplement(s) for figure 4:

**Figure supplement 1.** Expression of Tau-GFP in excitatory neurons in macaque brain.

**Figure supplement 2.** Representative cortical projections of vlPFC.

**Figure supplement 3.** The spatial extent of axonal clusters in the frontal cortex.

**Figure supplement 4.** Representative subcortical projections of vlPFC.

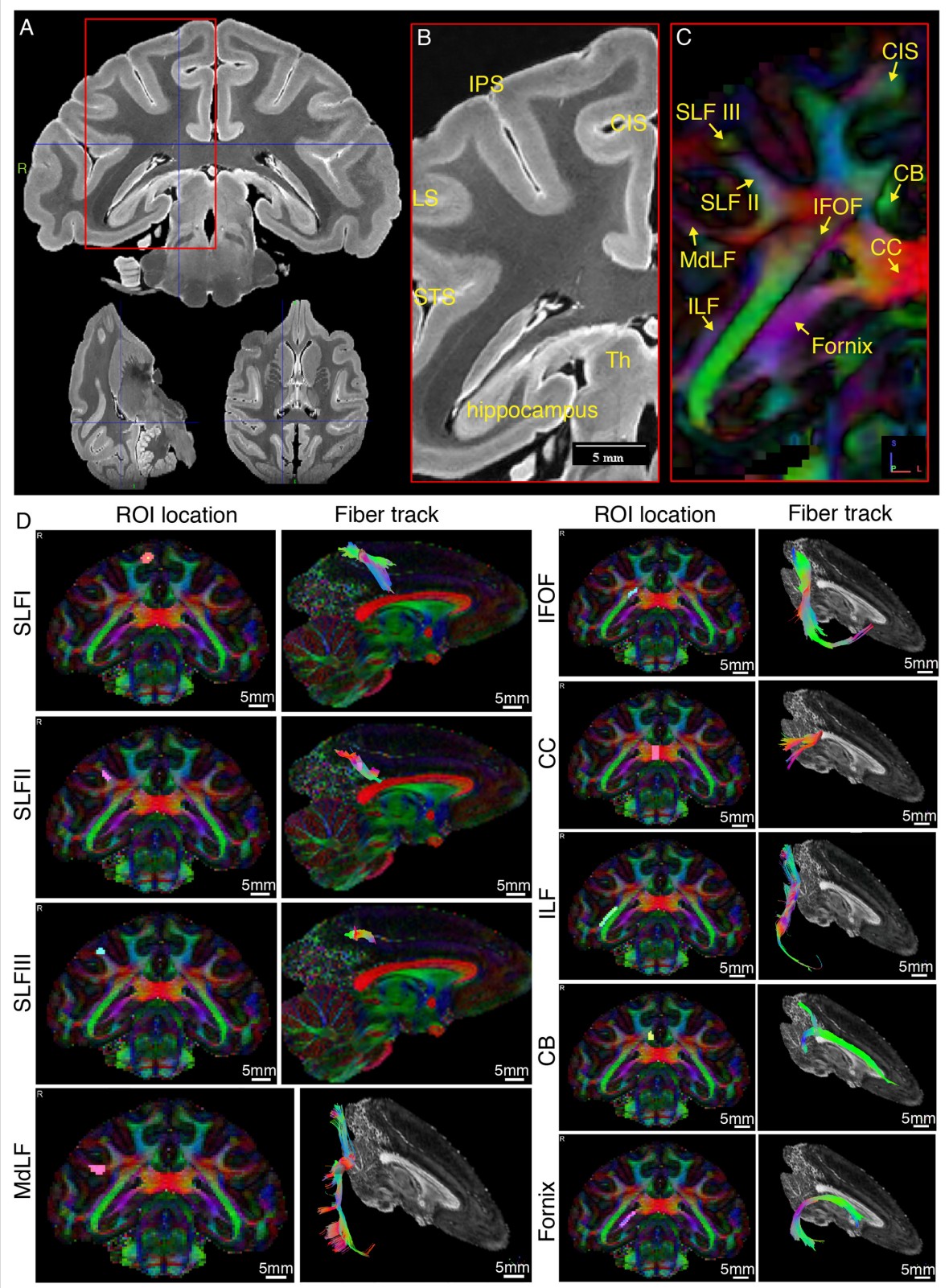

**Figure 5.** Representative *ex-vivo* MRI images of macaque brain. (**A**) Typical high-resolution T2-weighted images were shown in axial, coronal, and sagittal planes. (**B**) Zoom-in view of the red box in panel A, shown with anatomical landmark gyri including intraparietal sulcus (IPS), lunate sulcus (LS), superior temporal sulcus (STS), and cingulate sulcus (CIS). (**C**) The color-coded FA map corresponding to B. Major fiber bundles including superior longitudinal fasciculus subcomponent I, II, and III (SLF-I, -II, -III), inferior fronto-occipital fasciculus (IFOF), inferior longitudinal fasciculus (ILF), middle

*Figure 5 continued on next page*

*Figure 5 continued*

longitudinal fasciculus (MdLF), corpus callosum (CC), cingulum bundle (CB), and fornix are clearly demonstrated. Red color codes left and right, blue color codes superior and inferior, and green color codes anterior and posterior directions. (**D**) Typical tractography of the main fiber bundles indicated in C are derived from the present dMRI data. The ROI locations and fiber tracks are overlaid on the color-coded FA maps.

by 500 µm) along vlPFC-CC-contralateral tract, we observed consistent and significant correlations between these two modalities (*R* = 0.4624 ± 0.0922; Dice = 0.4312 ± 0.0861). Two example GFP-labeled axon images as marked in *Figure 6F* were displayed in *Figure 6G–J* with different magnifications, showing typical traveling axons in corpus callosum (*Figure 6G and H*) and frontal white matter (*Figure 6I and J*).

## Inferior fronto-occipital fasciculus in macaques

As illustrated by diffusion tractography, the inferior fronto-occipital fasciculus (IFOF) in macaques is a long-ranged bowtie-shaped tract (*Figure 7A*), showing traveling course similar to humans. The frontal stem of IFOF spread to form a thin sheet, and its temporal stem narrowed in coronal section, mainly gathered at the external capsule. The intersection between IFOF and axonal projections of vlPFC was shown in a common 3D space of diffusion tractography in *Figure 7B*, whereby the posterior part of vlPFC axonal projections apparently ends at the middle superior temporal region, far from the occipital lobe. To quantify the spatial correspondence between the IFOF tract and vlPFC projectome, the Szymkiewicz-Simpson overlap coefficient was calculated in a shared common 3D space after co-registration. It was 0.0397 and 0.0535 in 3D space for two dMRI data sets, indicating that only a small fraction of the IFOF tract and vlPFC projectome overlapped (mainly in the front half of the brain, *Figure 7*). Also, the Szymkiewicz-Simpson overlap coefficients between 2D coronal slices of IFOF and vlPFC projectome were plotted along the anterior-posterior axis of the macaque brain (*Figure 7C*). The anterior part of the vlPFC axonal projections shown by STP tomography largely overlapped with the dMRI-derived IFOF tracts in frontal whiter matter (*Figure 7D*), external capsule (*Figure 7E*), claustrum (*Figure 7E and F*), and extreme capsule (*Figure 7F*). Meanwhile, the posterior part of dMRI-derived IFOF tract passed through temporal white matter (*Figure 7G*), whereas the posterior part of fiber projections of vlPFC sent no axons to this region (*Figure 7G*).

We observed SLFIII linking the inferior parietal lobe to frontal lobe by traveling horizontally through the white matter in the macaque brain (*Figure 7—figure supplement 1A*). The spatial trajectory of SLFIII derived from dMRI tractography was validated through the comparison of virally labeled axonal fibers. Most axonal fibers of the SLFIII tract travel perpendicular to the coronal plane (cutting plane) such that these labeled fibers appeared mainly as green dots in high magnification fluorescent images (*Figure 7—figure supplement 1B-G*). Similarly, we calculated the Szymkiewicz-Simpson overlap coefficients between the SLFIII derived from diffusion tractography and the parietal branch of vlPFC projections identified from viral tracing in two brain samples, as showed markedly high overlap (*r* = 0.2603 and 0.2175). Unlike the IFOF in which the results of diffusion tractography mainly overlapped with those of viral tracing in the anterior part (*Figure 7*), the dMRI-derived SLFIII substantially overlapped with the vlPFC axonal projections in frontal and parietal whiter matter along the whole trajectory (*Figure 7—figure supplement 1*).

## Discussion

### Brain-wide excitatory projectome of vlPFC in macaques

We customized STP tomography for whole-brain imaging of the macaque monkey at submicron resolution and accomplished brain-wide 3D reconstruction of axonal connectome, thanks to prominent characteristics of STP tomography including free of tissue distortions, no need for section-to-section alignment, and high-resolution image sets readily warped in 3D space (*Amato et al., 2016*). Importantly, we coupled STP tomography with genetic methods using enhancers/promoter elements that target specific cell types (*Luo et al., 2018*). Previous studies have demonstrated that a CaMKIIα promoter carried by lentivirus was able to target excitatory neurons with optogenetic proteins in the macaque brain (*Han et al., 2009*), and a TH promoter carried by AAV selectively targeted dopamine neurons (*Stauffer et al., 2016*). Here we deployed AAV with CaMKIIα promoter to focus on the

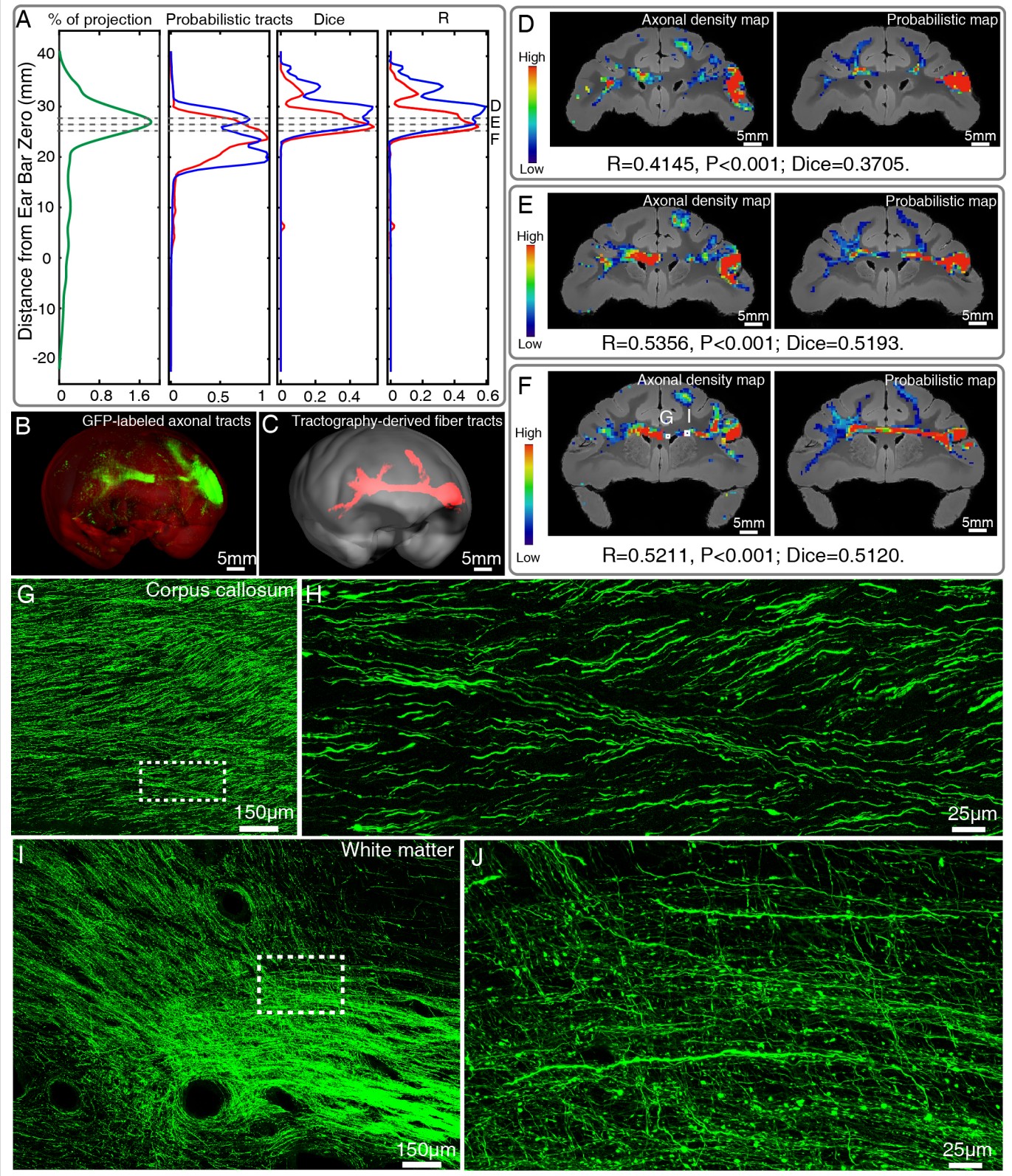

**Figure 6.** Comparison of ventrolateral prefrontal cortex (vlPFC) connectivity profiles by serial two-photon (STP) tomography and diffusion tractography. (**A**) Percentage of projection, Probabilistic tracts, Dice coefficients, and Pearson coefficients (**R**) were plotted along the anterior-posterior axis in the macaque brain. Blue and red colors indicate results of two dMRI data sets acquired from different macaque monkeys. (**B, C**) 3D visualization of the fiber tracts issued from the injection site in vlPFC to corpus callosum to the contralateral vlPFC by STP tomography and diffusion tractography.

*Figure 6 continued on next page*

*Figure 6 continued*

(**D–F**) Representative coronal slices of the diffusion tractography map and the axonal density map along the vlPFC-CC-contralateral tract, overlaid with the corresponding anatomical MR images. (**G–J**) GFP-labeled axon images as marked in (**F**) were shown with magnified views. (**H, J**) correspond to high magnification images of the white boxes indicated in G and I, both of which presented a great deal of details about axonal morphology.

excitatory projection of vlPFC, whereby immunofluorescent staining with both CaMKIIα and GABA confirmed that GFP was specifically expressed in excitatory neurons. Hence, this integrated approach allows clear dissection of projection patterns from diverse neuronal types (***Bedbrook et al., 2018***), and enriches our knowledge about the anatomical infrastructure of neural circuits for individual cell types at the entire brain scale (***Stauffer et al., 2016***).

Both anterograde and retrograde tracing evidence shows that vlPFC is extensively connected to other divisions of PFC including OFC, FEF, and the ventral premotor cortex. Extrinsic connections beyond the PFC, vlPFC is connected mainly to the dysgranular insula, frontal operculum, somatosensory-related areas in the parietal operculum and inferior parietal cortex, visual-related areas in the inferior temporal cortex, and anterior cingulate areas (***Saleem et al., 2014***; ***Gerbella et al., 2010***). We found the excitatory projection of vlPFC to the rest of brain was compatible with previous reports using chemical tracers (***Borra et al., 2011***; ***Gerbella et al., 2010***; ***Safadi et al., 2018***). Furthermore, we compared the current vlPFC projectome data with the well-known macaque connectivity database CoCoMac (***Bakker et al., 2012***), which includes the results of several hundred published axonal tract-tracing studies in the macaque monkey brain (***Stephan, 2013***). Essentially the vlPFC connectivity profile shown here was markedly similar to that of CoCoMac, except that vlPFC projections to PFCol, PFCdm, PFCdl, PFCoi, PFCm, PMCvl, amygdala, and SII have not been reported in CoCoMac database or reported merely with unspecified strength. In addition, we compared vlPFC projections with one recent report (***Gerbella et al., 2010***), showing that the brain regions projected from area 45 were clearly observed in the present vlPFC projection data. One recent study compared results of terminal labeling using Synaptophysin-EGFP-expressing AAV (specifically labeling synaptic endings) with the cytoplasmic EGFP AAV (labeling axon fibers and synaptic endings). There was high correspondence between synaptic EGFP and cytoplasmic EGFP signals in target regions (***Oh et al., 2014***). Thus, we relied on quantifying GFP-positive pixels (containing signals from both axonal fibers and terminals) rather than the number of synaptic terminals, similarly done in recent reports (***Oh et al., 2014***; ***Gehrlach et al., 2020***).

## AAV2/9 is suitable for long-range axonal tracing in the macaque brain

Methods for tissue labeling have been continuously evolving from silver impregnation of degenerating fibers to ex-vivo visualization of axonally transported tracers injected at single brain nuclei, and finally to an integrated style which coupled high-resolution whole-brain imaging technologies with viral and genetic tracers (***Nassi et al., 2015***). Among four viral vectors tested here, we found that AAV2/9 demonstrated the highest efficiency of long-range axonal tracing in the macaque brain. VSV was initially utilized as a transsynaptic tracer in a previous study since VSV encodes five genes, including G protein which promotes anterograde transsynaptic spread among neurons (***Beier et al., 2011***). In our study, we used VSV with G deletions to trace axonal projection without trans-synaptic labeling, which enabled robust gene expression at remarkably higher level relative to other vectors in a very short time (less than a week). But we found that a shorter expression time of VSV-ΔG was insufficient to label axons traveling long distance whereas a longer expression time of VSV-ΔG caused cell death, consistent with a prior finding that VSV-G failed to label transsynaptic cells at distant areas (***Mundell et al., 2015***). The advantage of lentivirus, which is derived from human immunodeficiency virus type 1 (HIV-1) (***Naldini et al., 1996***), is that it has a large genetic capacity of approximately 10 Kb which allows for the expression of multiple gene and usage of more than one promoter or regulatory elements. And we found GFP expression induced by lentivirus remarkably stable after 9 months in macaque monkeys, even though the labeled level was mild (***Schambach et al., 2013***) and the labeled scope was limited.

As an effective carrier for gene delivery into the brain, AAV has a number of established advantages including minimal toxicity, weak host immune response, stable gene expression in neurons with extraordinarily high transfection efficiency (titers up to $10^{12}$–$10^{13}$ genome copies per mL) (***Bedbrook***

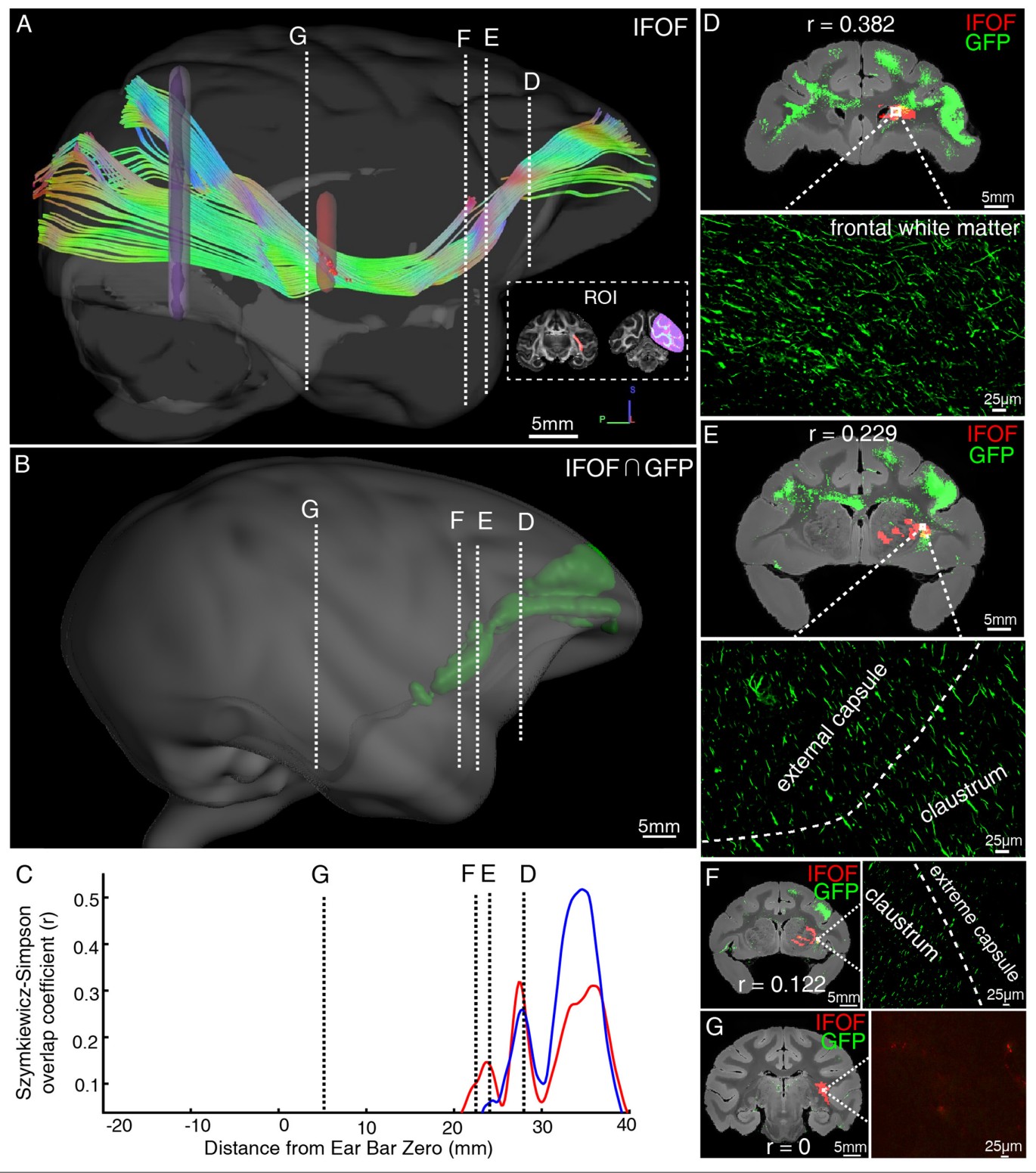

**Figure 7.** Illustration of the inferior fronto-occipital fasciculus by diffusion tractography and serial two-photon (STP) tomography. (**A**) The fiber tractography of inferior fronto-occipital fasciculus (IFOF) (lateral view). Two inclusion ROIs at the external capsule (pink) and the anterior border of the occipital lobe (purple) were used and shown on the coronal plane. The IFOF stems from the frontal lobe, travels along the lateral border of the caudate nucleus and external/extreme capsule, forms a bowtie-like pattern and anchors into the occipital lobe. (**B**) The reconstructed traveling course of IFOF based on vlPFC projectome was shown in 3D space. (**C**) The Szymkiewicz-Simpson overlap coefficients between 2D coronal brain slices of the dMRI-derived IFOF tract and vlPFC projections were plotted along the anterior-posterior axis of the macaque brain. Blue and red colors indicate results

*Figure 7 continued on next page*

*Figure 7 continued*

of two dMRI data sets acquired from different macaque monkeys. Four cross-sectional slices (**D–G**) along the IFOF tracts were arbitrarily chosen to demonstrate the spatial correspondence between the diffusion tractography and axonal tracing of STP images. (**D–G**) The detected GFP signals (green) of vlPFC projectome and the IFOF tracts (red) obtained by diffusion tractography were overlaid on anatomical MRI images, with a magnified view of the box area. Evidently, there was no fluorescent signal detected in the superior temporal area where the dMRI-derived IFOF tract passes through (**G**).

The online version of this article includes the following figure supplement(s) for figure 7:

**Figure supplement 1.** Illustration of SLFIII by diffusion tractography and STP tomography (segmented out of the vlPFC-parietal projections).

---

*et al., 2018*). One major drawback of AAV vectors is the limited packaging capacity. AAVs usually deliver gene cassettes of approximately 4.8 Kb (i.e. one or two small genes) (*Nassi et al., 2015*), which has motivated us in pursuit of biocompatible nano-based carriers (*Cui et al., 2019*). It is well known that different AAV serotypes have their own sequences in the inverted terminal repeats such that they have distinct transfection bases toward various cell types in the brain. The recombinant virus we used was AAV2/9 which contains the inverted terminal repeats from AAV serotype 2 and the capsid proteins from AAV serotype 9. Previous studies have shown that AAV2 is the most widely used AAV vector and effectively transfects neurons of nonhuman primates (*Watakabe et al., 2015*). In a recent report on a mouse model, researchers co-injected AAV and a classical antegrade tracer - biotinylated dextran amine (BDA) into one brain region and observed long-range projections with similar patterns by both tracers, except that BDA had more retrograde-labeled neurons, probably uptaken by passing fibers in some areas (*Oh et al., 2014*). Together, our results have demonstrated that AAV2/9 vector was more suitable for long-range axonal fiber tracing, while VSV-ΔG was suitable for rapid gene expression and lentivirus for long-term gene expression in macaques.

## Comparison of STP tomography with diffusion tractography

Pioneering studies on cross-modality comparison across the whole-brain scale have been done by constructing a connectivity matrix using dMRI-based tractography and tracer-injection tracing in mice (*Calabrese et al., 2015*) and in monkeys (*Donahue et al., 2016*; *Girard et al., 2020*; *van den Heuvel et al., 2015*). The spatial correspondence of axonal fibers derived from diffusion tractography and GFP-labeled fluorescent images have been compared both in mice (*Harsan et al., 2013*; *Chen et al., 2015*; *Chang et al., 2017*) and in macaques (*Dauguet et al., 2007*). Dauguet and coworkers found that the somatosensory and motor tracts derived from diffusion tractography were visually in good agreement with the reconstructed 3D histological sections labeled by anterograde WGA-HRP tracer in a monkey brain, but suffered certain limitations for regions at remote locations from seeds (*Dauguet et al., 2007*). Moreover, the structural connectivity analyses based on the histological dataset provided varying correlative evidence between these two measurements (like $r = 0.21$ *van den Heuvel et al., 2015* using the CoCoMac tracer data *Stephan, 2013* and $r = 0.59$ *Donahue et al., 2016* using the tracer connectivity matrix from *Markov et al., 2014*). Note that such structural connectivity analysis does not describe a 3D correspondence of the axonal fiber trajectory, but an 'end-to-end' match. STP tomography effectively transformed a series of histological slice images into a 3D space with which dMRI-derived tracts were co-registered, thus enabling a direct, quantitative comparison of the high-throughput data from these two modalities. This is technically challenging due to a giant difference in scale between the axonal fibers and image resolution of dMRI (*Glasser et al., 2016*). We have taken meticulous steps to maximize the signal-to-noise ratio like using Gd-DTPA as an enhanced contrast agent (*D'Arceuil et al., 2007*) and to minimize the image artifacts in an ultrahigh field scanner for achieving a reasonably high spatial resolution. We observed that GFP-labeled axonal density maps not only significantly overlapped with dMRI-derived probabilistic maps throughout the traveling course but also demonstrated comparable connectivity strengths and patterns. But caution should be born in mind that diffusion tractography estimated the Brownian motion of water molecules, from which the directionality of axons cannot be distinguished (*Mori and Zhang, 2006*). The viral tracing data here contained only anterograde axonal fiber projections.

Our particular focus on the vlPFC connectivity profile leads us to clarify the existence of the IFOF in monkeys which is heavily debated (*Forkel et al., 2014*). The IFOF in human brain was first described in the early 20th century (*Curran, 1909*), whereby the anatomy of this pathway in human has been recently shown by micro-dissection and diffusion tractography studies (*Hau et al., 2016*; *Takemura et al., 2017*). Its entire course through the ventral part of the external capsule (EC) connects the occipital cortex and

the parietal and temporal cortices to the frontal cortex (*Sarubbo et al., 2019*). Some axonal tracing studies showed connections between frontal and occipital lobes in monkeys (*Gerbella et al., 2010*; *Markov et al., 2014*), which was consistent with the observation by tractography (*Feng et al., 2017*) and blunt dissection (*Sarubbo et al., 2019*) experiments. By contrast, other studies that are capable of tracking monosynaptic pathways failed (*Petrides, 2013*). Using the same ROIs seeds as prior studies (*Barrett et al., 2020*; *Feng et al., 2017*), our ex-vivo tractography did show fiber connections between frontal and occipital lobes in monkeys, matching the trajectory of IFOF in humans. By contrast, using anterograde AAV vector without trans-synaptic capability, we found that vlPFC fiber projections passed through external capsule, claustrum, and extreme capsule and anchored to the middle superior temporal region. Although the trajectory of vlPFC between frontal and temporal regions matched well with the diffusion tractography of IFOF, axonal projections of vlPFC never reached the occipital lobe. Lack of monosynaptic tracing data in human subjects, we could not rule out the possibility of same scenario for IFOF in humans. If the IFOF connects the frontal lobe with the occipital lobe in a trans-synaptic manner, it unveils a hitherto unknown information relay/integration process occurring in superior temporal area of the primate species which holds great implications for neural network computation. Nevertheless, unlike the direct monosynaptic connections reported between subdivisions of PFC such as OFC and the visual cortex in mice (*Zingg et al., 2014*; *Liu et al., 2020*), our results underscore a nontrivial species difference and raise interesting questions about the long-range brain organization and the functional role of superior temporal area in primates which definitely merits future examination.

In summary, we present a detailed excitatory connectivity projection map from vlPFC to the entire macaque brain, and demonstrate a broadly applicable roadmap of integrating 3D STP tomography labeled with antero-/retro-grade tracer and diffusion tractography for the mesoscopic mapping of brain circuits in the primate species.

# Materials and methods

## Key resources table

| Reagent type (species) or resource | Designation | Source or reference | Identifiers | Additional information |
|---|---|---|---|---|
| Antibody | anti-CaMKIIa (Rabbit polyclonal) | Abcam | Cat# ab5683, RRID: AB_305050 | IF(1: 200) |
| Antibody | anti-GABA (Rabbit polyclonal) | Abcam | Cat# ab8891, RRID: AB_306844 | IF(1: 200) |
| Antibody | anti- NeuN (Mouse Monoclonal) | Millipore | Cat# MAB377, RRID: AB_2298772 | IF(1: 500) |
| Antibody | anti- GFP (Rabbit polyclonal) | Thermo Fisher Scientific | Cat# A-11122, RRID: AB_221569 | IF(1: 300) |
| Antibody | anti- GFAP (Rabbit polyclonal) | Boster (PMID:31101714) | Cat# PB0046 | IF(1: 200) |
| Antibody | Goat anti-Rabbit IgG (H + L) Alexa Fluor 405 (Goat polyclonal) | Thermo Fisher Scientific | Cat# A31556, RRID: AB_221605 | IF(1: 500) |
| Antibody | Donkey anti-Rabbit IgG (H + L) Alexa Fluor 568 (Donkey polyclonal) | Thermo Fisher Scientific | Cat# A10042, RRID: AB_2534017 | IF(1: 600) |
| Antibody | Goat anti-Mouse IgG (H + L) Alexa Fluor 568 (Goat polyclonal) | Thermo Fisher Scientific | Cat# A11031, RRID: AB_144696 | IF(1: 600) |
| Antibody | Goat anti-Mouse IgG (H + L) Alexa Fluor Plus 647 (Goat Polyclonal) | Thermo Fisher Scientific | Cat# A32728, RRID: AB_2633277 | IF(1: 500) |
| Antibody | Goat anti-Rabbit IgG (H + L) Alexa Fluor 488 (Goat Polyclonal) | Thermo Fisher Scientific | Cat# A11034, RRID: AB_2576217 | IF(1: 300) |
| Software, algorithm | Fiji (ImageJ) software | NIH | RRID:SCR_002285 | http://fiji.sc |
| Software, algorithm | MATLAB | Mathworks | RRID:SCR_001622 | https://www.mathworks.com/ |
| Software, algorithm | FSL software | University of Oxford | RRID:SCR_002823 | http://www.fmrib.ox.ac.uk/fsl |
| Software, algorithm | ANTS - Advanced Normalization ToolS | University of Pennsylvania | RRID:SCR_004757 | http://www.picsl.upenn.edu/ANTS/ |

## Animals and ethics statement

All experimental procedures for nonhuman primate research in this study were approved by the Animal Care Committee of Shanghai Institutes for Biological Sciences, Chinese Academy of Sciences, and conformed to the National Institutes of Health guidelines for the humane care and use of laboratory

**Table 1.** Injection cases and viral vectors used in this study.

| ID | Species | Injection site | Viral vector | Expression time |
|---|---|---|---|---|
| #1 | Macaca mulatta (M) | Cortex<br>MD | VSV-ΔG-Tau-GFP<br>VSV-ΔG-Tau-GFP | ~5 d<br>~5 d |
| #2 | Macaca mulatta (M) | Cortex<br>MD | VSV-ΔG-Tau-GFP<br>VSV-ΔG-Tau-GFP | ~30 d<br>~90 d |
| #3 | Macaca fascicularis (M) | Cortex<br>Cortex<br>Cortex<br>MD | AAV2/9-CaMKIIα-Tau-GFP<br>AAV2/9-hSyn- mCherry<br>Lenti-UbC-GFP<br>Lenti-UbC-GFP | ~45 d<br>~45 d<br>~260 d<br>~260 d |
| #4 | Macaca fascicularis (M) | MD | AAV2/9-CaMKIIα-Tau-GFP | ~60 d |
| #5 | Macaca fascicularis (M) | MD | AAV2/9-CaMKIIα-Tau-GFP | ~60 d |
| #6 | Macaca fascicularis (F) | MD | AAV2/9-CaMKIIα-Tau-GFP | ~60 d |
| #7 | Macaca fascicularis (M) | vlPFC | AAV2/9-CaMKIIα-Tau-GFP | ~75 d |
| #8 | Macaca fascicularis (F) | vlPFC | AAV2/9-CaMKIIα-Tau-GFP | ~75 d |
| #9 | Macaca fascicularis (F) | vlPFC | AAV2/9-CaMKIIα-Tau-GFP | ~110 d |
| #10 | Macaca fascicularis (F) | \ | \ | \ |

Abbreviations: M, male; F, female. MD, mediodorsal thalamus; vlPFC, ventrolateral prefrontal cortex; UbC, human ubiquitin C; hSyn, human synapsin I; CaMKII, Ca2+/calmodulin dependent protein kinase II; d, day.

animals. From November 2015 till November 2020, ten adult macaque monkeys (Macaca mulatta and Macaca fascicularis) weighting 3.5–12.2 kg (6.94 ± 2.98 kg) were used for in this study (**Table 1**), two of which (Macaca fascicularis) were used for ex-vivo ultrahigh field dMRI scanning.

## Viral vectors

Four viral vectors, including VSV-ΔG (VSV-ΔG-Tau-GFP, titer: $5.0 \times 10^8$ PFU/mL), lentivirus (lentivirus-UbC-GFP, titer: $1.33 \times 10^9$ TU/mL), and two constructs of AAV2/9 (AAV2/9-CaMKIIα-Tau-GFP, titer: $8.47 \times 10^{13}$ vg/mL; AAV2/9-hSyn-mCherry, titer: $1.6 \times 10^{13}$ vg/mL), were tested in this study (**Table 1**). AAV2/9 and VSV-ΔG were purchased from BrainVTA technology Co., Ltd. (Wuhan, China), and lentivirus was provided by a coauthor Z.Q's laboratory. Here, the recombinant AAV2/9 contained either a hSyn or CaMKIIα promoter to regulate the expression of either reporter gene mCherry in all neurons or fused Tau-GFP protein in glutamatergic excitatory neurons, respectively. Regarding the VSV vector, G protein was deleted to prevent transsynaptic spread. The last tested viral vector, Ubic promoter-driven lentivirus, expressed GFP in all eukaryotic cells.

## MRI-Guided Virus Injection

To precisely target brain regions in individual subjects, we performed in-vivo MRI scanning in monkeys and then used MRI images to guide the virus injection. T1 weighted images for each monkey were obtained with a 3T scanner (Siemens Tim Trio, Erlangen, Germany) under general anesthesia. A detailed description of in-vivo MRI scanning procedure has been described in our previous studies (**Cai et al., 2020**; **Zhan et al., 2021**; **Wang et al., 2013**; **Lv et al., 2021**; **Lv et al., 2016**; **Zhang et al., 2019**) and briefly summarized here. Anesthesia was induced by intramuscular injection of ketamine (10 mg per kg). Deep anesthesia was maintained by isoflurane (1.5–3%) and vital physiological signals were continuously monitored during MRI scanning. Anatomical scans were acquired with an MPRAGE sequence using the following parameters: TR = 2300ms, TE = 2.8ms, TI = 1100ms, spatial resolution 0.5 mm isotropic. The target regions were localized in each animal by warping the 3D digital atlas of Saleem and Logothetis (**Reveley et al., 2017**) to the individual T1 image using a symmetric normalization (SyN) algorithm. The location of the vlPFC was then calculated with regard to the stereotaxic space.

All procedures for virus injection were performed in strict aseptic conditions. The head of the animal was fixed in a stereotaxic apparatus, within the same coordinate space as the MRI images. The target area was then labeled and an incision was made to expose the skull. A burr hole with a 2 mm radius was drilled above the target according to the calculated coordinates, and the dura was carefully

incised to expose the cortical surface. The viral vector was delivered into the cortex using a 33-gauge Hamilton syringe controlled by an UltraMicroPump and a micro4 controller (WPI). The injection speed started with 200 nl/min and was increased to 400 nl/min; total injection volume was 10–20 µl. After injection, the needle was retained for at least 15 min and drawn back at a rate of ~1 mm/min. The burr hole was then filled with bone wax and the skin was sutured. Cephalosporin was given for three consecutive days after surgery (25 mg/kg/day, i.m., once a day).

## Cryo-Sectioning

According to the expression time of individual virus (*Table 1*), animals were deeply anesthetized, and then transcardially perfused with 0.9% NaCl (pH = 7.2) followed by ice-cold 4% paraformaldehyde in 0.01 M phosphate buffered saline. Brains were extracted and post-fixed in 4% PFA for 3 days. Cryo-sectioning combined with wide field microscope imaging and confocal laser microscope imaging was performed for virus testing. The fixed brain was first cut into a block, then equilibrated sequentially in 15 and 30% sucrose in PBS until it sank to the bottom of the container. A cryostat microtome (Leica CM1950) was used to serially slice the brain into 50 µm sections. Brain slices were preserved in a cryoprotectant solution (containing 30% ethanediol, 30% sucrose in PBS solution, pH = 7.2) for further immunofluorescence staining and imaging.

## Serial Two-Photon Tomography

Fluorescence signals of AAV labeled areas were detected and recorded using a customized STP tomography (*Figure 1—figure supplement 1*). To image the monkey brain, we customized the STP tomography system which was integrated a two-photon microscope (Bruker) with a vibratome (WPI) (*Figure 1—figure supplement 1*), computer controlled and fully automated. The XY stage covered a 50*60 mm$^2$ area, and the 3D scanning of Z-volume stacks was achieved with using a stepper motor (Thorlabs) that traveled over 70 mm. The fixed brain that was embedded with 4% agarose was scanned in a 3T MRI to obtain ex-vivo T1 images. Using these T1 images as reference, the active imaged region of each section was determined during STP tomography for improved imaging efficiency. The embedded brain was then held via a magnetic adaptor to a stepper motor and immersed in a cutting bath filled with PBS containing 0.1% sodium azide. The vibratome blade was aligned in parallel with the leading edge of the specimen block. Brain images were captured from the anterior PFC to posterior V1 in the coronal plane. Fluorescence signals for the green channel (excitation wavelength light in 920 nm) and red channel (excitation wavelength light in 1045 nm) were acquired at 30 µm below the cutting surface through a Nikon 16 x Water objective (N.A. = 0.8).

During serial scanning, the STP tomography system was fully automated: each optical section was imaged as a mosaic of fields of view on the block surface as the xy stage moved the brain under the objective; once an entire section was imaged, the xy stage moved the brain to the vibratome and cut off a 200 µm section from the top of the sample. The remaining specimen was then moved back under the objective for imaging the next neighboring plane. Optical and mechanical sectioning were repeated until the complete brain data was collected. Hence fluorescent images of the whole monkey brain were continuously acquired (*Figure 1—figure supplement 2*).

High x-y resolution (0.95 µm/pixel) serial 2D images were acquired in the coronal plane at a z-interval of 200 µm across the entire macaque brain. The scanning time of a single field-of-view which contains 1,024 by 1,024 pixels was 1.629 s (i.e. pixel residence time was ~1.6 µs), as resulted in a continuous ~1 month scanning and ~5 TB STP tomography data for a single monkey brain (*Figure 1—figure supplement 2*). Once finished scanning, all sections were retrieved from the cutting bath and stored in cryo protection solution (containing 30% glycol, 30% sucrose in PBS) at –20°C for further histological examination. Three samples were injected with AAV in vlPFC, and two of them were able to be imaged with STP tomography. Unfortunately, one sample became 'loose' and fell off from the agar block after several weeks of imaging. So, the quantitative results were not shown in *Figure 3*.

## Histological staining

To perform immunofluorescence procedure, brain slices were incubated in blocking solution containing 5% BSA and 0.3% Triton X-100 in PBS at room temperature for 2 hr and then overnight with primary antibodies in PBS containing 3% BSA and 0.3% Triton X-100 at 4 °C. Slices were rinsed in PBS followed by Alexa Fluor-conjugated secondary antibodies at room temperature for 3 hr, and DAPI

(Cell signaling Cat# 4083 s) for 30 min at room temperature. The following primary antibodies were used: CaMKIIa (1:200, Abcam, Cat# ab5683, RRID: AB_305050), GABA (1:200, Abcam Cat# ab8891, RRID:AB_306844), NeuN (1:500, Millipore, Cat# MAB377, RRID:AB_2298772), GFP (1:300, Thermo Fisher Scientific, Cat# A-11122, RRID:AB_221569), GFAP (1:200, Boster, Cat# PB0046). The following secondary antibodies were used: Goat anti-Rabbit IgG (H + L) Alexa Fluor 405 (1:500, Thermo Fisher Scientific, Cat# A31556, RRID:AB_221605), Donkey anti-Rabbit IgG (H + L) Alexa Fluor 568 (1:600, Thermo Fisher Scientific, Cat# A10042, RRID:AB_2534017), Goat anti-Mouse IgG (H + L) Alexa Fluor 568 (1:600, Thermo Fisher Scientific, Cat# A11031, RRID:AB_144696), Goat anti-Mouse IgG (H + L) Alexa Fluor Plus 647 (1:500, Thermo Fisher Scientific, Cat# A32728, RRID:AB_2633277), Goat anti-Rabbit IgG (H + L) Alexa Fluor 488 (1:300, Thermo Fisher Scientific, Cat# A11034, RRID:AB_2576217). The brain slices were mounted onto customized 2 × 3 inch or 3 × 4 inch glass slides. The sections were then scanned using an Olympus VS120 (Olympus, Japan), a wide field microscope, with a U Plan Super Apo 10 × objective (N.A. = 0.4) at a resolution of 0.65 μm/pixel. High resolution fluorescent images were acquired with a confocal laser microscope Nikon TiE (Nikon, Tokyo, Japan) with a Plan Fluo 40 × Oil DIC N2 objective (N.A. = 1.3), 0.5 μm Z-interval, and 1024 × 1,024 pixels.

## Fluorescence image preprocessing

Fluorescent images of the macaque brain usually contain strong autofluorescence signal (*Figure 1—figure supplement 3A-E*), mainly caused by the accumulation of lipofuscin (*Economo et al., 2016*). Autofluorescence provides good contrast between gray matter and white matter, which is rather useful for image registration. But the presence of autofluorescence is undesirable for the axon tracing procedure since this background signal sometimes is much stronger than that of some thin GFP labeled axons (*Figure 1—figure supplement 3A-E*). Nevertheless, thanks to the broad emission spectrum of lipofuscin (*Hunnicutt et al., 2016*), autofluorescence and GFP signals are easily distinguishable from each other. We therefore implemented and compared the following three methods for background reduction: (1) transforming the GFP signal from the green channel (488 nm) to the blue channel (405 nm) using immunofluorescent staining (*Figure 1—figure supplement 3M*), (2) subtracting the normalized autofluorescence signal in the red channel from the green channel (*Figure 1—figure supplement 3F*), which contains both GFP signal and autofluorescence background signal, (3) supervised machine learning for autofluorescence exclusion (*Figure 1—figure supplement 3J*).

The first method involved staining the brain tissue with anti-GFP antibody and Alexa Fluor 405 conjugated secondary antibody to transform the GFP signal from a green channel to a blue channel. Unlike the green and red channels, the transferred blue channel (*Figure 1—figure supplement 3M*) did not contain high-intensity autofluorescence puncta. Although this post-hoc thick-section immunofluorescent method successfully reduced autofluorescence, it was incompatible with the block face imaging method. The second one was to subtract the normalized red channel from the green channel using the broad emission spectrum characteristic of autofluorescence puncta, which was able to remove high intensity background signal (*Figure 1—figure supplement 3F*). The third was based on a supervised machine learning plugin for ImageJ, trainable WEKA segmentation (*Arganda-Carreras et al., 2017*), which classifies and binarizes GFP and autofluorescence background signal for background exclusion (*Figure 1—figure supplement 3J*). Both subtraction and machine learning methods were used for better visualization of fluorescence images when necessary, whereas only the supervised machine learning approach was used for quantitative analysis of STP data (*Hunnicutt et al., 2016*).

## STP Image Processing

STP tomography data processing included axonal fiber detection, image stitching, down sampling, cross-modality registration and quantification. The data analysis was undertaken on a compute cluster with a 3.1–3.3 GHz 248 core CPU, 2.8T of RAM, and 17472 CUDA cores. Fluorescent images of primate (*Abe et al., 2017*) brain often contain high-intensity dot-looking background signal caused by accumulation of lipofuscin. Thanks to the broad emission spectrum of lipofuscin, dot-looking background and GFP-positive axonal varicosities are easily distinguishable from each other. For instance (*Figure 1—figure supplement 4*), axonal varicosities can be selectively excited in green channel, while dot-looking background lipofuscin usually present in both green channel and red channel. During quantitative analysis, a machine learning algorithm was adopted to reliably segment the GFP labeled axonal fibers including axonal varicosities, and remove the lipofuscin background (*Arganda-Carreras*

*et al., 2017*; *Gehrlach et al., 2020*). The total computational time for the machine learning predictions in one macaque brain was ~1.5 months. To evaluate overall classifier performance, the precision–recall F measure, also called F-score, was computed by using additional four labeled images as test sets. Higher accuracy performance achieved by the classifier often yield higher F-scores (94.41 ± 1.99%, mean ± S.E.M.). During STP tomography scanning, each field of view (FOV) was saved as a 1024 × 1024-pixel image. For image stitching, individual FOV images from red channel, green channel and segmented GFP signal were stitched into full tissue sections using the Terastitcher software. A convolutional neural network-based denoising approach was used to improve SNR of images when necessary (*Krull et al., 2019*). The natural alignment of serial images generated by STP tomography allowed to stack the section images to form a coherent reconstructed 3D volume (*Ragan et al., 2012*). In order to localize the virus injection site, a threshold was set at green channel to retain the fluorescence signal only from the cell soma for each section image. Images of the red channel and injection site volumes were downsampled to a resolution of 200 × 200 × 200 μm grid. For serial segmented GFP images, the total signal intensity was computed for each 200 × 200 μm grid by summing the number of signal-positive pixels in that voxel. Red channel volume was used to perform registration to the monkey brain template, as red channel images contain visible anatomical information of brain structures (*Kuan et al., 2015*). The brain template of cynomolgus macaque was adopted from an MRI-based atlas generated from 162 cynomolgus monkeys (*Lv et al., 2021*). We warped the red channel volume to the template space by using a symmetric normalization (SyN) algorithm in ANTs (*Figure 1—figure supplement 5*). For registration to the 3D common space, it took half an hour approximately. The cortical label was adopted from the D99 parcellation map (*Reveley et al., 2017*), and subcortical label was adopted from INIA19 parcellation map (*Rohlfing et al., 2012*). Also the segmented GFP volume and injection site volume were co-registered onto the same template. Density of GFP signal and total GFP volume in each parcellated brain region were used to represent the axonal connectivity strength. Percent of total projection was defined by the GFP-positive pixel count within each parcellated brain region (or brain lobe) normalized to the total of all GFP-positive pixels. Additionally, the percent innervation density was calculated as the proportion of density of GFP pixel counts covering the maximal density of GFP pixel counts of the brain. To create plots that display the data along the anterior-posterior axis (e.g. % density innervation), the location of ear bar zero was used as the origin. The percent innervation density of each cortical region innervated by vlPFC was rendered onto a brain surface. *Figures 3–4* and *Figure 4—figure supplements 2–4* were derived from sample #8 with infected area in 45, 12 l and 44 of vlPFC. *Figure 1—figure supplement 6* was derived from sample #7 with infected area in 12 l and 45 of vlPFC.

## Ex-Vivo MRI Scanning and Data Preprocessing

We collected dMRI data using an 11.7T horizontal MRI system (Bruker Biospec 117/16 USR, Ettlingen, Germany), equipped with a 72 mm volume resonator and an actively shielded, high performance BGA-S series gradient system (gradient strength: 740 mT/m, slew rate: 6660 T/m/s). After a fixation period of ~30 days, the whole brain specimen was immersed in a 1:100 dilution of a 1 mmol/mL gadolinium MR contrast agent (Magnevist, Bayer Pharma AG, Germany) mixed with phosphate buffered saline (PBS) solution for 14 days. Before MRI scanning, the specimen was washed and drained of water from the surface, then positioned into a customized container which was 3D printed for perfect accommodation of the brain sample. Thus the brain was held steadily during MRI scanning. And the container was filled with FOMBLIN perfluoropolyether (Solvay Solexis Inc Thorofare, NJ, USA) for susceptibility matching and improved magnetic field homogeneity. The specimen was degassed with a vacuum pump for 24 h under 0.1 atmosphere pressure to remove all air bubbles in the sample at 20 °C (magnet room temperature). The ex-vivo macaque brain was scanned on a 11.7T animal MRI system (Bruker Biospec 117/16 USR, Ettlingen, Germany), equipped with a 72 mm volume resonator and an actively shielded, high performance BGA-S series gradient system (gradient strength:740 mT/m, slew rate: 6660 T/m/s). dMRI images were acquired using a 3D diffusion-weighted spin echo pulse sequence with single-line read-out, TR/TE = 82/22.19ms, FOV = 64 × 54 mm, matrix = 128 × 108, slice thickness = 0.5 mm and averages = 3, which included 60 diffusion directions with b = 4000 s/mm$^2$ ($\Delta/\delta$ = 15/2.8ms, maximum b value = 4234.97,, gradient amplitude = 97.19 mT/m) and five nondiffusion encoding (b = 0 s/mm$^2$) directions. For the ex-vivo diffusion MRI data acquisition, the b-value was recommended to set at 4000 s/mm$^2$ (*D'Arceuil et al., 2007*; *Dyrby et al., 2011*). T2

weighted images were acquired using a 2D Turbo RARE sequence with TR/TE = 8353.42/28.8ms, flip angle = 87°, matrix = 450 × 450, FOV = 54 × 45 mm, slice thickness = 0.5 mm, and averages = 6. T1 weighted images were acquired using 3D FLASH sequence with TR/TE = 40/5.5ms, flip angle = 15°, matrix = 290 × 225, FOV = 58 × 45 mm, slice thickness = 0.2 mm, and averages = 4. All scanning was performed at room temperature (approximately 20 °C) and the total scan time was approximately 36 hr.

Visual inspection of MRI data was first performed to ensure that there were no obvious image artifacts and geometric distortions. Then we calculated the signal-to-noise ratio (SNR) for typical diffusion images. As diffusion images were acquired by spin warp imaging (image reconstruction by a 3D Fourier transform) with a volume quadrature coil, the SNRs were calculated using the 'two-region' approach (*Dietrich et al., 2007*; *Kaufman et al., 1989*). Specifically, for each gradient encoding direction, the deep white matter (WM) were extracted in subject-native diffusion space to represent the signal (*Sijbers et al., 1998*); a region positioned in the no signal area at the corner of the image was used to represent the noise. As a rule of thumb, the SNR of b = 0 s/mm$^2$ images should be minimally larger than 20 for obtaining relatively unbiased measures of parameters such as FA (*Mukherjee et al., 2008*). Typical SNRs of diffusion images with b = 0 and b = 4000 in the present study were 48.34 ± 8.50 and 23.13 ± 2.05, respectively. It allowed a reliable seed-based 3D reconstruction for diffusion tractography, as illustrated in *Figure 5*. The dMRI data was preprocessed using the FSL software (http://www.fmrib.ox.ac.uk/fsl) (*Behrens et al., 2007*). Individual image volumes were co-registered with b = 0 images to account for eddy currents and B0 drift using affine registration in FLIRT (*Jenkinson and Smith, 2001*). A custom in-house script was applied to reorient the corresponding gradient direction matrix. Careful steps have been taken to minimize artifacts caused by motion and field distortion, and image correction was applied only if necessary (*Andersson and Sotiropoulos, 2016*; *Andersson and Sotiropoulos, 2015*).

## Reconstruction and comparison of diffusion tractography and axonal tracing

We first identified the injection-site volume in the vlPFC in STP tomography data, warped it to the space of dMRI volume and used it as a seed mask for tractography. Then the injection site related tractography was constructed using the preprocessed dMRI images in FSL toolbox. BEDPOSTX was used for Bayesian estimation of a crossing fiber model with three-fiber orientation structure for each voxel using Markov chain-Monte Carlo sampling (*Behrens et al., 2007*). This provided a voxel-wise estimate of the angular distribution of local tract direction for each fiber, which was a starting point for tractography. Tractography was then performed from the injection-site seed masks without waypoint mask and termination mask using the Probtrackx probabilistic tractography software (*Behrens et al., 2007*). A probabilistic map of fiber tracts was generated with 500 µm isotropic resolution. A probabilistic map provided, at each voxel, a connectivity value, corresponding to the total number of samples that passed from the seed region through that voxel. The following settings were used: number of samples per voxel = 5000, number of steps per sample = 2000, step length = 0.2 mm, loop check, default curvature threshold = 0.2 (corresponding to a minimum angle of approximately ±80 degrees), subsidiary fiber volume threshold = 0.01, seed sphere sampling = 0 and no way-point or termination mask. In the resulting map, each voxel's value represented the degree of connectivity between it and the seed voxels. To generate dMRI-derived fiber tracts, the resulting probabilistic maps were set at a threshold, i.e., any voxel below threshold was set to zero. In parallel, segmented fluorescence images from the STP tomography data were downsampled to 500 µm grid to generate axonal density maps. The signal intensity of an axonal density map was computed for each 500 × 500 µm$^2$ grid by summing the number of GFP-positive pixels within that area. Note that the axonal density map was also filtered by setting an intensity threshold of $10^{2.8}$ to minimize false positives due to segmentation artefacts (*Oh et al., 2014*). After co-registered the probabilistic maps and the axonal density maps onto the same template, both Dice coefficients and pixel-wise Pearson coefficients were calculated to quantitatively assess the spatial overlap (*Dice, 1945*; *Crum et al., 2006*).

As described recently (*Barrett et al., 2020*), the inferior fronto-occipital fasciculus was reconstructed using streamline-based probabilistic tractography. We ran this probabilistic tractography tool in MRTrix3 (https://www.mrtrix.org/) via bootstrapping (*Jones, 2008*). Streamlines were seeded over the whole brain area that encapsulated the tract of interest. Two inclusion masks were used to define

two regions that each tract must pass through, and only streamlines that pass through both regions are retained. One exclusion mask was used to restrict tracking to the contralateral hemisphere of the brain. The inclusion and exclusion masks were drawn manually as described previously (*Barrett et al., 2020*): the first mask was placed on the anterior border of the occipital lobe in the coronal view, the second mask was placed on the external/extreme capsules in the coronal view, and the third mask was cover the whole left hemisphere as the exclusion mask. The delineation process was performed using the MRIcro (https://crnl.readthedocs.io/) software. Using this 'waypoint' method, the resultant streamlines were able to meet our preset conditions. To further reduce false positive tracts, any streamlines that were identified as either attached to other tracts or anatomically implausible trajectories were manually removed.

According to previous studies (*Sani et al., 2019*; *de Schotten et al., 2011*), two inclusion ROIs and two exclusion ROIs were used for reconstructing the SLFIII in diffusion tractography. Two inclusion regions were placed in the frontal and parietal lobes (violet and red areas marked in *Figure 7—figure supplement 1A*). Two exclusion regions were placed in the temporal and occipital lobes, respectively.

Details of streamline-based probabilistic tractography processing were described here. The fiber orientation distribution function (FOD) was estimated with MRtrix3 (https://www.mrtrix.org/) (*Tournier et al., 2012*) using the *tournier* algorithm for single-tissue Constrained spherical deconvolution *Tournier et al., 2013*. For fiber tracking, we then used *tckgen* with the *Tensor_Prob* tracking algorithm in MRtrix3 (*Jones, 2008*). Within each image voxel, a residual bootstrap was performed to obtain a unique realization of the dMRI data in that voxel for each streamline. These data are then sampled via trilinear interpolation at each streamline step, the diffusion tensor model is fitted, and the streamline follows the orientation of the principal eigenvector of that tensor. The following additional tckgen settings and inputs were used: step size of 0.25 mm, max. angle between successive steps = 45°, max. length = 150 mm, min. length value set the min. length 10 mm, cutoff FA value = 0.1, b-vectors and b-values from the diffusion-weighted gradient scheme in the FSL format, b-value scaling mode = true, maximum number of fibers = 10,000, and unidirectional tracking.

For a direct comparison between diffusion-derived the IFOF tract and vlPFC projection fibers, we first generated the track density images of the IFOF tract and co-registered them onto the space of the template. The spatial overlap of GFP-positive vlPFC projection fibers and the IFOF tract were then detected with using ImageJ and FSL software in both 2D and 3D space (*Figure 7B*). The Szymkiewicz-Simpson overlap coefficient was adopted to quantify the spatial relationship between the IFOF tract and vlPFC projectome, which was defined as the size of the union of them over the size of the smaller set:

$$\text{overlap (IFOF, vlPFC)} = \frac{|\text{IFOF} \cap \text{vlPFC}|}{\min(|\text{IFOF}|, |\text{vlPFC}|)}$$

The Szymkiewicz-Simpson overlap coefficient ranges from 0 (no overlap) to 1 (if the IFOF tract is found in its entirety in vlPFC projectome).

## Materials availability

All the virus vectors used in this paper are available from the authors for sample test.

## Acknowledgements

We would like to thank Jinqiang Peng and Jie Xu for their assistance to data acquisition, and thank Drs. John Gore, Ed Callaway and Anna Roe for their stimulating discussions and suggestions during the preparation of this study. This work was supported by the Key-Area Research and Development Program of Guangdong Province (2019B030335001), National Natural Science Foundation (No. 82151303), National Key R&D Program of China (No. 2021ZD0204002), Shanghai Municipal Science and Technology Major Project (No. 2018SHZDZX05), the Strategic Priority Research Program of Chinese Academy of Science (No. XDB32000000), and Peking-Tsinghua Center for Life Sciences.

## Additional information

### Funding

| Funder | Grant reference number | Author |
| --- | --- | --- |
| Key-Area Research and Development Program of Guangdong Province | 2019B030335001 | Zheng Wang |
| National Key Research and Development Program of China | No. 2021ZD0204002 | Zheng Wang |
| Chinese Academy of Sciences | Strategic Priority Research Program No. XDB32000000 | Zheng Wang |
| National Natural Science Foundation of China | 82151303 | Zheng Wang |
| Shanghai Municipal Science and Technology Commission | Major Project No. 2018SHZDZX05 | Zheng Wang |
| Peking-Tsinghua Center for Life Sciences | | Zheng Wang |

The funders had no role in study design, data collection and interpretation, or the decision to submit the work for publication.

### Author contributions

Mingchao Yan, Wenwen Yu, Data curation, Formal analysis, Investigation, Methodology, Writing – original draft; Qian Lv, Data curation, Formal analysis, Investigation, Methodology, Validation; Qiming Lv, Tingting Bo, Xiaoyu Chen, Yilin Liu, Yafeng Zhan, Shengyao Yan, Xiangyu Shen, Formal analysis, Investigation, Methodology; Baofeng Yang, Investigation, Methodology; Qiming Hu, Jiangli Yu, Data curation, Formal analysis, Methodology; Zilong Qiu, Yuanjing Feng, Xiao-Yong Zhang, He Wang, Conceptualization, Methodology, Resources; Fuqiang Xu, Conceptualization, Investigation, Methodology, Resources; Zheng Wang, Conceptualization, Funding acquisition, Investigation, Project administration, Supervision, Writing – review and editing

### Author ORCIDs

Tingting Bo http://orcid.org/0000-0002-9080-5165
Xiao-Yong Zhang http://orcid.org/0000-0001-8965-1077
He Wang http://orcid.org/0000-0002-2053-9439
Fuqiang Xu http://orcid.org/0000-0002-4382-9797
Zheng Wang http://orcid.org/0000-0001-7138-8581

### Ethics

All experimental procedures for nonhuman primate research in this study were approved by the Animal Care Committee of Shanghai Institutes for Biological Sciences, Chinese Academy of Sciences (ER-SIBS-221601P), and conformed to the National Institutes of Health guidelines for the humane care and use of laboratory animals.

### Decision letter and Author response

Decision letter https://doi.org/10.7554/eLife.72534.sa1
Author response https://doi.org/10.7554/eLife.72534.sa2

## Additional files

### Supplementary files

• Transparent reporting form

## Data availability

There is no publicly accessible resource for hosting such big connectome data. Therefore we host it ourselves on an institutional FTP server which can be accessed via username and password (available upon request). We commit to keeping it available for at least 5 years, and provide alternative procedures where users can copy any or all of it to their own computer if needed.

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
