## [Editor Report]

This paper uses a novel technique in combination with high throughput microscopy to generate a detailed map of macaque prefrontal connections. It will not only be of interest to anatomists, but also to the neuroimaging community, as it includes a detailed comparison to MRI-based connectivity approaches commonly used to study the human brain.

---

## [Decision Letter]

**Decision letter after peer review:**

Thank you for submitting your article "Mapping brain-wide excitatory projectome of primate prefrontal cortex at submicron resolution: relevance to diffusion tractography" for consideration by *eLife*. Your article has been reviewed by 2 peer reviewers, and the evaluation has been overseen by a Reviewing Editor and Timothy Behrens as the Senior Editor. The following individual involved in review of your submission has agreed to reveal their identity: Pavel Osten (Reviewer #1).

Essential revisions:

Most major concerns are on the quantification of the tracer data. Please address these comments raised by both reviewers.

*Reviewer #2 (Recommendations for the authors):*

The introduction of this manuscript can be improved to help the reader better understand the background of the IFOF debate and why it is important. Schmahmann and Pandya have made an excellent review of the history of this debate in their book Fiber Pathways of the Brain, Chapters 18 and 19, which may be helpful.

[Editors' note: further revisions were suggested prior to acceptance, as described below.]

Thank you for resubmitting your work entitled "Mapping brain-wide excitatory projectome of primate prefrontal cortex at submicron resolution and comparison with diffusion tractography" for further consideration by *eLife*. Your revised article has been evaluated by Timothy Behrens (Senior Editor) and a Reviewing Editor.

The manuscript has been improved but there are some remaining issues that need to be addressed, as outlined below:

Whilst the reviewer is happy with the changes to the paper, there is one last thing they are asking for which the reviewing editor agrees with.

*Reviewer #2 (Recommendations for the authors):*

The authors have provided substantial supplemental material and revised text in response to my previous comments. The revised manuscript has shown solid improvement. However, there are two remaining issues regarding the insufficient data to support the study's conclusions.

The main concern is that the authors have not adequately addressed comment #3, which is directly related with the cross-species dishomology conclusion. Perhaps it helps to reiterate the rationale of the previous comment for clarity. The authors compared the tracing and tractography results of the vlPFC-CC-contralateral tract in one animal (Figure 6) and the IFOF in another (Figure 7). Based on the consistency in the vlPFC-CC-contralateral tract, the authors concluded that the inconsistency in the IFOF reveals an artifact in tractography. However, the inconsistency could also be caused by individual differences or tracer issues such as weak transport. To rule out the individual difference, cross-monkey variability should be examined, such as showing the same tract in both monkeys. To rule out tracer issues, the authors need to demonstrate that at least a majority of the vlPFC fiber tracts show consistency between tracing and tractography. The reason provided by the authors (insufficient resolution in the z-dimension thus can only compare within the coronal plane) cannot argue against comparing additional tracts, because: (1) The IFOF runs in the anterior-posterior direction rather than the coronal plane. If only within-plane comparison is valid, how would one justify the comparison for IFOF? (2) There are ways to go around the resolution issue such as comparing 3D reconstructed bundles with streamline-volumes instead of pixel-to-pixel comparison. After all, the 200-micron resolution in the z-dimension is sufficient for 3D reconstruction of any major tract.

A related remaining issue is the visualization of tract tracing results, mentioned in the original comment #2.3. The authors did not provide coronal plots that are comparable to the traditional visualization in the tract tracing literature. These plots should contain complete coronal slices showing representative labeling patterns in the gray and white matter (see e.g. Figure 4 in Petrides and Pandya, 2007, J Neurosci). The rationale is that such a depiction would help validate the tracing results with previous findings. The added Figure 4 supplement 2 contains close-up depictions of cortical areas in parts, which is not helpful for assessing the global labeling pattern, and the 25 micron close-ups are unnecessary.

Perhaps a potential resolution to both issues is to show the labeling in SLF III and compare the tracing and tractography results of this bundle. The vlPFC is known to have dense fibers in SLF III, which runs roughly parallel and superior to IFOF in the anterior-posterior axis (see e.g. Figure 18 in Frey, Mackey and Petrides, 2014, Brain and Language). If the authors can show that tracing and tractography consistently capture SLF III, it would be a strong support to the difference found in IFOF.

---

## [Author Response]

Essential revisions:Most major concerns are on the quantification of the tracer data. Please address these comments raised by both reviewers.Reviewer #2 (Recommendations for the authors):The introduction of this manuscript can be improved to help the reader better understand the background of the IFOF debate and why it is important. Schmahmann and Pandya have made an excellent review of the history of this debate in their book Fiber Pathways of the Brain, Chapters 18 and 19, which may be helpful.

We thank the reviewer for the thoughtful suggestion. We have added the following description into the “Introduction” section as follows:

“The IFOF first proposed in the early 19th century supposedly connects the ventrolateral prefrontal cortex and medial orbitofrontal cortex to the occipital lobe through the ventral part of the external capsule (Curran, 1909; Catani et al., 2002). Micro-dissection and diffusion MRI tractography studies have recently confirmed the anatomy of this pathway (Sarubbo et al., 2019; Barrett et al., 2020). Despite an abundance of functional evidence supporting a central role of occipito-frontal circuitry in cognition and sensory integration, a number of axonal tracing studies, which have been able to identify monosynaptic connections, have failed to reveal the IFOF in the macaque brain (Schmahmann and Pandya, 2006; Schmahmann et al., 2007). By contrast, sparse connections between frontal and occipital cortices in macaques were reported by other labs (Gerbella et al., 2010; Markov et al., 2014), although they do not show whether these axons follow the course expected for the IFOF. As such, a detailed anatomical definition of the IFOF is still under debate (Barrett et al., 2020).”

[Editors' note: further revisions were suggested prior to acceptance, as described below.]

Reviewer #2 (Recommendations for the authors):The authors have provided substantial supplemental material and revised text in response to my previous comments. The revised manuscript has shown solid improvement. However, there are two remaining issues regarding the insufficient data to support the study's conclusions.The main concern is that the authors have not adequately addressed comment #3, which is directly related with the cross-species dishomology conclusion. Perhaps it helps to reiterate the rationale of the previous comment for clarity. The authors compared the tracing and tractography results of the vlPFC-CC-contralateral tract in one animal (Figure 6) and the IFOF in another (Figure 7). Based on the consistency in the vlPFC-CC-contralateral tract, the authors concluded that the inconsistency in the IFOF reveals an artifact in tractography. However, the inconsistency could also be caused by individual differences or tracer issues such as weak transport. To rule out the individual difference, cross-monkey variability should be examined, such as showing the same tract in both monkeys. To rule out tracer issues, the authors need to demonstrate that at least a majority of the vlPFC fiber tracts show consistency between tracing and tractography. The reason provided by the authors (insufficient resolution in the z-dimension thus can only compare within the coronal plane) cannot argue against comparing additional tracts, because: (1) The IFOF runs in the anterior-posterior direction rather than the coronal plane. If only within-plane comparison is valid, how would one justify the comparison for IFOF? (2) There are ways to go around the resolution issue such as comparing 3D reconstructed bundles with streamline-volumes instead of pixel-to-pixel comparison. After all, the 200-micron resolution in the z-dimension is sufficient for 3D reconstruction of any major tract.

We apologize for causing some misunderstanding about the data demonstration. The comparisons of vlPFC and IFOF tracts between two modalities were conducted one-to-one in one monkey and repeated in another monkey. In order to show results from both monkeys, we used blue and red colors to represent two different monkeys in Figure 6A and Figure 7C. To demonstrate the methodological consistency between viral tracing and tractography, we conducted pixelwise comparison throughout the whole brain in coronal plane from anterior to posterior including other projections (Figure 6A).

As the Reviewer suggested, we further reconstructed the SLFIII by using ultra-high field diffusion MRI data from two macaque monkeys, and compared with the viral tracing data, shown in a new supplementary Figure (Figure 7—figure supplement 1). The description of the “Methods and Materials” and “Results” sections has been revised accordingly in the revision.

“According to previous studies (Thiebaut de Schotten et al., 2011; Sani et al., 2019), two inclusion ROIs and two exclusion ROIs were used for reconstructing the SLFIII in diffusion tractography. Two inclusion regions were placed in the frontal and parietal lobes (violet and red areas marked in Figure 7—figure supplement 1A). Two exclusion regions were placed in the temporal and occipital lobes, respectively.”

“We observed SLFIII linking the inferior parietal lobe to frontal lobe by travelling horizontally through the white matter in the macaque brain (Figure 7—figure supplement 1A). The spatial trajectory of SLFIII derived from dMRI tractography was validated through the comparison of virally labeled axonal fibers. Most axonal fibers of SLFIII tract travel perpendicular to the coronal plane (cutting plane) such that these labeled fibers appeared mainly as green dots in high magnification fluorescent images (Figure 7—figure supplement 1 B-G). Similarly, we calculated the Szymkiewicz-Simpson overlap coefficients between the SLFIII derived from diffusion tractography and the parietal branch of vlPFC projections identified from viral tracing, as showed markedly high overlap (r = 0.2603 and 0.2175). Unlike the IFOF of which the results of diffusion tractography mainly overlapped with those of viral tracing in the anterior part (Figure 7), the dMRI-derived SLFIII substantially overlapped with the vlPFC axonal projections in frontal and parietal whiter matter along the whole trajectory (Figure 7—figure supplement 1).”

A related remaining issue is the visualization of tract tracing results, mentioned in the original comment #2.3. The authors did not provide coronal plots that are comparable to the traditional visualization in the tract tracing literature. These plots should contain complete coronal slices showing representative labeling patterns in the gray and white matter (see e.g. Figure 4 in Petrides and Pandya, 2007, J Neurosci). The rationale is that such a depiction would help validate the tracing results with previous findings. The added Figure 4 supplement 2 contains close-up depictions of cortical areas in parts, which is not helpful for assessing the global labeling pattern, and the 25 micron close-ups are unnecessary.

We thank the reviewer for the helpful suggestion. As suggested, we have added a new figure showing the coronal slices as Supplementary materials in the revised manuscript.

Perhaps a potential resolution to both issues is to show the labeling in SLF III and compare the tracing and tractography results of this bundle. The vlPFC is known to have dense fibers in SLF III, which runs roughly parallel and superior to IFOF in the anterior-posterior axis (see e.g. Figure 18 in Frey, Mackey and Petrides, 2014, Brain and Language). If the authors can show that tracing and tractography consistently capture SLF III, it would be a strong support to the difference found in IFOF.

We thank the reviewer for his/her constructive suggestion on the validation of SLF III tract. We have conducted additional analysis and provided a new figure showing SLF III (Figure 7—figure supplement 1). Relevant references have been included in the revision.